# LEARNING TO REASON ACROSS PARALLEL SAMPLES FOR LLM REASONING

## ABSTRACT

Scaling test-time compute brings substantial performance gains for large language models (LLMs). By sampling multiple answers and heuristically aggregate their answers (e.g., either through majority voting or using verifiers to rank the answers), one can achieve consistent performance gains in math domains. In this paper, we propose a new way to leverage such multiple sample set. We train a compact LLM, called Sample Set Aggregator (SSA), that takes a concatenated sequence of multiple samples and output the final answer, optimizing it for the answer accuracy with reinforcement learning. Experiments on five reasoning datasets demonstrate both the efficacy and efficiency of SSA. Notably, SSA improves over naive majority voting by 8% pass@5 on MATH. Furthermore, our 3B SSA surpasses model-based re-ranking with a much larger 72B process reward model. Our analysis also shows promising generalization ability of SSA, across sample set sizes, base model families and scales, and tasks. By separating LLMs to generate answers and LLMs to analyze and aggregate sampled answers, our approach can work with the outputs from premier black box models easily and efficiently.

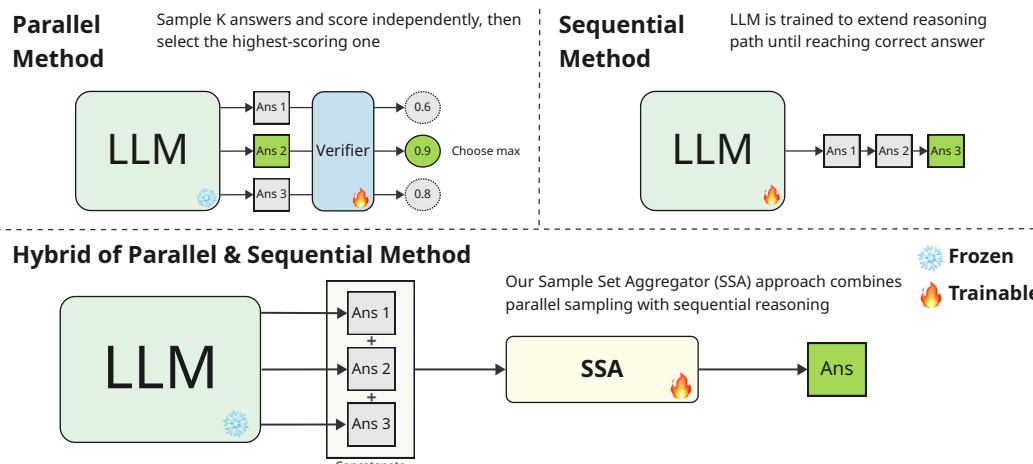

Figure 1: Illustration of our approach (bottom), parallel method (top left), and sequential method (top right). We train a compact LLM, called Sample Set Aggregator (SSA), to take a concatenated sequence of multiple samples and output the final answer.

## 1 INTRODUCTION

Recent advances in large language models (LLMs) have significantly enhanced their ability to perform complex reasoning tasks (El-Kishky et al., 2024; DeepSeek-AI et al., 2025). Orthogonal to approaches to improve training LLMs through better learning objectives (Ouyang et al., 2022; Rafailov et al., 2023), architectures (Gu & Dao, 2024; Peng et al., 2023) or training dataset composition (Gunasekar et al., 2023; Wettig et al., 2025), recent work (Snell et al., 2025) explores a new dimension: test-time scaling. While having the LLM fixed, by allocating more computation at inference time (e.g., through repeated sampling and majority voting), one can improve the final task performance.

In general, test-time scaling methods fall into two paradigms. Parallel scaling generates multiple reasoning paths independently and aggregates them via strategies such as majority voting or best-of-N selection (Wang et al., 2023; Uesato et al., 2022; Lightman et al., 2024). Sequential scaling, on the other hand, iteratively refines a single solution, often through prompting-based self-reflection or by incentivizing iterative computation (DeepSeek-AI et al., 2025; Muennighoff et al., 2025; Kumar et al., 2025). In this paper, we introduce a novel test-time scaling approach that leverages both parallel and sequential scaling. Figure 1 illustrates our approach in comparison with prior work.

We separately have an LM to draft multiple answers and another LM to combine multiple answers to generate the final answer, naming the latter Sample Set Aggregator (SSA). SSA is optimized with reinforcement learning (RL) to maximize final answer accuracy. Unlike parallel scaling approaches that mostly view individual samples from LLMs in isolation, SSA can interpret multiple generations as representations of the LM's output distribution, thus directly optimizing the synthesis of the final answer based on the landscape of the output distribution.

We conduct extensive experiments across five reasoning benchmarks under controlled test-time compute budgets. Results show that SSA substantially narrows the gap between actual model performance and oracle-best accuracy (pass@K), outperforming standard parallel strategies such as reward-based reranking. Moreover, we demonstrate that a compact SSA model can match the performance of reinforcement-trained larger models used in sequential scaling, suggesting the effectiveness of SSA as a lightweight way for sequential scaling. Further analysis highlights the generalization capabilities of SSA: SSA trained on one dataset for a particular model can successfully synthesize outputs from different model families and sizes across different tasks.

We summarize our key contributions and findings as follows:

- *SSA*, a lightweight LLM is introduced that concatenates $K$ parallel candidates from a frozen base model and then performs one sequential RL step to generate the final answer. This single pass unifies the strengths of parallel and sequential test-time scaling, showing strong performance gain while training only a small model.
- Conceptually, we propose to *reason over the output distribution.* Specifically, instead of training the base LM, SSA optimizes over its sampled outputs. The base LM that produces answers remains to be a black box; SSA is trained only with sampled answers from base LMs.
- We observe broad and consistent empirical gains across five math benchmarks, two LLM families (Qwen 2.5, Llama 3.1) and three base sizes (7B, 14B, 32B), over strong baselines.

## 2 RELATED WORK

**Scaling test-time compute in parallel.** Recent research has established the effectiveness of increasing compute used at inference time, known as test-time scaling (El-Kishky et al., 2024; DeepSeek-AI et al., 2025; Snell et al., 2025; Brown et al., 2025). A prominent approach for test-time scaling focuses on parallel scaling, which samples multiple answers *independently* and aggregates them into a single answer. This aggregation can be performed through majority voting (Wang et al., 2023; 2024a) or more sophisticated selection mechanisms. For instance, some methods prompt language models to select from one of the multiple samples (Chen et al., 2024a), while others employ dedicated verifier models to score potential solutions (Cobbe et al., 2021; Uesato et al., 2022; Lightman et al., 2024; Li et al., 2023; Wang et al., 2024b) and take a weighted majority solution.

Beyond naively sampling multiple answers with fixed decoding strategy, researchers have explored advanced search strategies, such as beam search (Yao et al., 2023; Xie et al., 2023) and Monte-Carlo tree search (MCTS) (Li et al., 2025; Xie et al., 2024). These search-based methods typically rely on verifiers (Xie et al., 2023) or process reward models (Cobbe et al., 2021; Wang et al., 2024b) to guide the decoding process, while still evaluating different rollouts independently. In contrast to these approaches that primarily assess samples in isolation, our approach learns to compare different samples jointly to determine the final answer.

**Scaling test-time compute sequentially.** Another line of research focuses on sequential scaling, which increases compute by iteratively updating and refining a solution. One way is to prompt or train LLMs to self-refine their proposed solutions (Madaan et al., 2023; Kumar et al., 2025; Qu et al., 2024; Chen et al., 2024b) in an iterative manner. Recent work has demonstrated the possibility of incentivizing LLMs to spend more tokens in a single completion through reinforcement learning

(DeepSeek-AI et al., 2025; Team et al., 2025) or by forcing LLMs to continue their reasoning chains by appending "wait" token (Muennighoff et al., 2025). While sequential scaling often yields performance improvements, recent studies debate on whether it outperforms simpler parallel scaling when controlling for compute (Zeng et al., 2025b; Hochlehnert et al., 2025a;b). Our approach bridges these paradigms by operating on parallel samples but treating them as a sequence rather than as isolated instances. The growing length of CoT traces incurs substantial inference cost. Recent efforts have sought to improve efficiency by introducing length penalties (Aggarwal & Welleck, 2025; Sui et al., 2025), adaptive thinking (Fang et al., 2025; Zhang et al., 2025b; Lou et al., 2025), or applying early stopping (Zhang et al., 2025a; Yang et al., 2025b). In contrast, our approach trains a compact aggregator that achieves both strong performance and efficiency.

**Training language models for reasoning.** More broadly, our work aims to enhance LLM performance on reasoning tasks, which has been a central pursuit in LLM development. Substantial efforts have been devoted to training LLMs as reasoning policies (distributions more likely to contain correct answers) through supervised fine-tuning on collections of chain-of-thought (Azerbayev et al., 2024; Puerto et al., 2024; Luo et al., 2023), or through reinforcement learning with rewards on intermediate steps (Uesato et al., 2022; Wang et al., 2024b; Kazemnejad et al., 2025) or based solely on final answer correctness (DeepSeek-AI et al., 2025). Our research is more closely aligned with work on training LLMs as better verifiers for reasoning tasks and can complement solutions from any policy models. Existing methods train LLM verifiers to assess individual solutions using human annotations (Cobbe et al., 2021) or noisy labels derived from answer correctness (Wang et al., 2024b; Hosseini et al., 2024; Liang et al., 2024). Our approach differs in that we train LLMs to verify sequences of solutions using reinforcement learning with a verifiable reward signal, the final answer correctness.

# 3 METHOD

## 3.1 PROBLEM FORMULATION

Our method assumes two models: (1) $LM_{\text{ans}}$: a language model that generate a solution given an input, and (2) $LM_{\text{SSA}}$: a language model that takes an input and multiple solutions and generates the final solution.

Let $\mathbf{x}$ be an input problem token sequence drawn from a data distribution $\mathcal{D}$, our method solves the problem with the following two steps:

**Step 1: Parallel answer set generation.** In this step, we use the $LM_{\text{ans}}$, which defines a conditional distribution $LM_{\text{ans}}(y|\mathbf{x})$ over all possible tokens $y \in Y$, to sample $K$ candidate solution sequences from $LM_{\text{ans}}$, and obtain

$$Y_K = \{\mathbf{y_1}, \mathbf{y_2}, \ldots, \mathbf{y_K}\}, \quad y_i \sim LM_{\text{ans}}(\cdot \mid \mathbf{x}) \text{ independently.}$$

**Step 2: Analyzing parallel answer set for the final answer.** We use a separate trained language model $LM_{\text{SSA}}$ that takes sampled answer sequence as input to generate the final answer $\mathbf{y}_{\text{final}}$.

$$\mathbf{y}_{\text{final}} \sim LM_{\text{SSA}}(\cdot \mid \mathbf{x}, Y_K)$$

Crucially, we treat the $LM_{\text{ans}}$ as a black-box sampler of potential solutions, and train a much smaller model ($LM_{\text{SSA}}$) to do a post hoc aggregation as opposed to performing reinforcement learning on the answer model. Separating the answer generation model from the SSA introduces flexibility in choosing a different generation model, and enable the use of larger models that are difficult to fine-tune due to limited compute or are only accessible via APIs. Thus we essentially propose a general conceptual framework for test-time computing.

To enable an LLM to function as SSA $LM_{\text{SSA}}$ for synthesizing the final answer, we provide a natural language instruction directing the model to "think carefully and thoroughly evaluate the proposed answer, and identify one correct answer from the proposed candidates". Following common practice in prior work (DeepSeek-AI et al., 2025), we specify a structured output format, which we use to extract the final answer. Please refer to Appendix B Figure 8 for details of the prompt.

In the rest of this section, we will introduce how we train the $LM_{\text{SSA}}$ to maximize the correctness of its final answer. We consider two ways of optimizing the SSA: reinforcement learning (§ 3.2) and supervised-finetuning (§ 3.3).

## 3.2 Training SSA with Reinforcement Learning

Let $\mathbf{y}^*$ be the gold solution for $\mathbf{x}$. For the final $\mathbf{y}_{\text{final}}$ given by SSA $LM_{\text{SSA}}(\cdot \mid \mathbf{x}, Y_K)$, we design a verifiable reward $R(\mathbf{y}_{\text{final}}, \mathbf{y}^*)$ that mainly verifies the answer correctness, and update $LM_{\text{SSA}}$ to maximize the expected reward:

$$\mathbb{E}_{[(x,\mathbf{y}^*)\sim\mathcal{D},\, Y_K\sim LM_{\text{ans}}(\cdot\mid x)]}\big[R\big(LM_{\text{SSA}}(x,Y_K),\mathbf{y}^*\big)\big]. \quad R(\mathbf{y}_{\text{final}},\mathbf{y}^*) = \begin{cases} 1, & \text{if } \mathbf{y_i} \text{ is correct,} \\ 0.05, & \text{if only format is correct,} \\ 0, & \text{otherwise.} \end{cases}$$

Following prior work (DeepSeek-AI et al., 2025), our reward also consider the format of the output specified in the prompt. Specifically, we let the reward $R$ to be 1.0 if the answer is correct; we let the reward be 0.05 if the output follows the format and is incorrect; we let the reward to be 0.0 if the output does not follow and format.

We use the Group-Relative Policy Optimization (GRPO (Shao et al., 2024)) as our optimization algorithm, which simplified value function of PPO (Schulman et al., 2017) with a normalized reward from a group. For the convention of notation, we use $\pi_\theta$ for $LM_{\text{SSA}}$ with parameter $\theta$. For completeness, we describe it here. It maximizes:

$$J_{\text{GRPO}}(\theta) = \mathbb{E}_{\substack{\mathbf{x}\sim\mathcal{D} \\ Y_K\sim LM_{\text{ans}}(\cdot\mid\mathbf{x}) \\ \{\mathbf{y}_i\}\sim\pi_{\theta_{\text{old}}}(\cdot\mid\mathbf{x},Y_K)}} \frac{1}{G}\sum_{i=1}^{G}\frac{1}{|\mathbf{y}_i|}\sum_{t=1}^{|\mathbf{y}_i|}\Big[\min\big(\rho_{i,t}(\theta)\,\hat{A}_{i,t},\ \text{clip}\big(\rho_{i,t}(\theta),1-\varepsilon,1+\varepsilon\big)\hat{A}_{i,t}\big)\Big]$$
$$- \beta\, D_{\text{KL}}\big(\pi_\theta \,\|\, \pi_{\text{ref}}\big),$$

with importance weight $\rho_{i,t}(\theta) = \dfrac{\pi_\theta\big(\,y_{i,t}\mid\mathbf{x},Y_K,y_{i,<t}\big)}{\pi_{\theta_{\text{old}}}\big(\,y_{i,t}\mid\mathbf{x},Y_K,y_{i,<t}\big)}$ with the advantage as $\hat{A}_{i,t} = \frac{r_i - mean(\mathbf{r})}{std(\mathbf{r})}$, Reward as $r_i = R(\mathbf{y}_i,\mathbf{y}^*)$, $\mathbf{r} = r_1, r_2, ..., r_n$, and $\varepsilon, \beta$ are hyperparameters. This policy gradient method is to maximize the $\hat{A}_{i,t}$ which translates to expected reward from $r_i = R(\mathbf{y}_i,\mathbf{y}^*)$.

**Implementation** We use a training set combining GSM8K and MATH (Cobbe et al., 2021; Hendrycks et al., 2021). To prepare the dataset, we use Qwen2.5-7B-Instruct model as $LM_{\text{ans}}$ to generate five answers for each questions (Qwen et al., 2025). Then we concatenate the question and each answer (ordered randomly) as input context of SSA LLM. We filter out instances where the combined answers exceed 4000 tokens, or where fewer than three valid answers are available. This results a dataset with a total size of 17.4k. We set the GRPO sample size to be 8, resulting in $8*17.4k \approx 140k$ during training.

## 3.3 Training SSA With Supervised Finetuning

We also explore the training of the SSA via supervised fine-tuning (SFT), leveraging a stronger model to construct oracle reasoning paths across multiple sampled answers to identify correct final answer. Once we constructed the oracle dataset, we use the standard language modeling objective, but only training on the output part. We assume a dataset of oracle reasoning path that leads to $\mathbf{y}^*$. If $\mathbf{y}^* = (y_1^*, y_2^*, \ldots, y_T^*)$ is tokenized into $T$ tokens, the SFT loss is to minimize:

$$\mathcal{L}_{\text{SFT}}(\theta) = -\sum_{(\mathbf{x},Y_K,\mathbf{y}^*)\in\mathcal{D}}\sum_{t=1}^{T}\log\Big[LM_{\text{SSA}}\big(y_t^* \mid \mathbf{x}, Y_K, \mathbf{y}_{<t}^*\big)\Big],$$

**Implementation** We prompt Qwen 2.5 7B Instruct model for 5 candidate solutions per question in the GSM8K dataset. Then, we provide concatenated candidate solutions to GPT-4.1 Nano model along with the original question and ground-truth answer (The exact prompt is in Appendix B). GPT-4.1 Nano then provides a step-by-step reasoning process to identify and generate the best final answer from these candidates, yielding 7.47k training examples. Overall, this approach achieves a 96.24% match rate with the original ground-truth answers.[1]

---

[1]When we inspected the remaining cases, we find many ground-truth labels are incorrect.

# 4 EXPERIMENTS

## 4.1 EXPERIMENTAL SETTINGS

**Datasets.** For evaluation, we use an array of commonly used math reasoning datasets: the test split of GSM8K, MATH as the in-domain evaluation sets, and AIME 2024 (MAA, 2024), AMC 2023, and Olympiad (He et al., 2024) as the test sets. We use the extracted answers and grade them against the ground truth answers, using the library that has been used in prior work (Lightman et al., 2024).

**Base Models for Candidate Generation (LM$_{\text{ans}}$).** We use the Qwen-2.5-Instruct model, with sizes 7B, 14B, and 32B to generate $K$ of answers. We use a decoding temperature 0.5 to construct the training and test dataset. In the training phase, we use $k = 5$ to train the SSA model. In the testing phase, we evaluated the performance with $k = \{5, 10, 15\}$.

**Base Models for SSA (LM$_{\text{SSA}}$).** We use the Qwen-2.5-base model with sizes 0.5B, 1.5B, and 3B. We chose the Qwen-2.5 model family due to its popularity and wide availability for PRM verifiers, making it possible to compare against existing PRM verifiers from the same model family.

**Training Implementation Details.** For the training library, we use torchtune due to its efficient VRAM management (torchtune maintainers & contributors, 2024). We use GRPO batch sample group size 8, batch size 1, temperature 1.0, AdamW optimizer, KL coefficient 0.01, and learning rate 1e-5. We trained all experiments one epoch. For hardware, we use 8*H100 80GB for training. For shorter context, it is also possible to train with 48GB VRAM.

## 4.2 COMPARISON SYSTEMS

The following models are compared, including the proposed SSA with three variations.

**Rule-based Baselines.** We report pass@k, which reports the percentage of examples where any of $k$ number of solution is correct (Brown et al., 2025). Pass@k also serves as the oracle performance assuming we have an oracle verifier. We also report the **majority vote** (Wang et al., 2023), counting the most frequent answer among the answer set.

**Outcome Reward Model (ORM).** We use off the shelf Llama-3.1-8B ORM model trained with RLHF-Reward Modeling (Xiong et al., 2024) to re-rank multiple samples, selecting the best one scored by the reward. We will call it Llama-ORM (8B). It is trained with 273k data. For ORM evaluations, it adds an evaluation token at the end of the answer and the model will provide a score.

**Process Reward Model (PRM).** One prominent way to leverage multiple parallel sample is using process reward models. We use the Qwen-7B PRM (Qwen PRM) and Qwen-72B PRM model from Zhang et al. (2025c) to re-rank the candidate solutions.

We note that compared to the PRM model, our SSA is trained with significantly less resources regarding both model scale (0.5B - 3B vs 7B) and the amount of training data. Specifically, Qwen-7B PRM is trained with more than 500,000 queries with 6 to 8 answers with step labels, resulting over 3 million total training data (about $20\times$ larger than the training data size of SSA). For a more fair comparison, we also include Shepherd PRM (Wang et al., 2024b), 7B model trained with 440k total step level data from their MCTS roll out. Additionally, Qwen PRM initialized from Qwen2.5-Math-7B-Instruct (Yang et al., 2024), whereas we initialized from the Qwen 2.5 base model.

For evaluation, we follow the training method of Qwen PRM (Zhang et al., 2025c), and we separate steps with '\n\n' delimiter. Then we use PRM model to compute a score for each step and use their product to calculate the response score since this yielded the best performance in their experiments.

**Universal Self Consistency (USC) (Chen et al., 2024a).** USC is a prompting-based method takes a concatenation of the multiple parallel sample answers, and instructs a LM to generate the final answer. The exact prompt can be found in Appendix B. The task setting is equivalent to our SSA method, but the LM is not trained to optimize for the final answer.

**SSA and Varitions.** We train three types of the SSA models, one trained with the SFT objective only (SFT), one trained with SFT objective and then with the RL objective (SFT + RL), and one that is trained with RL objective only (RL). For SFT objective, we use the dataset described in Section 3.3 to

Table 1: Results (accuracy %) with $k = 5$ candidate answers generated by Qwen2.5-7B-Instruct as $LLM_{ans}$ model. Aggregation overheads (seconds/question) is measured end to end with AMC23 40 questions with k=5.

| Method | Aggregation Overhead (s) | Datasets | | | | | Avg |
|---|---|---|---|---|---|---|---|
| | | GSM8K | MATH | AIME24 | AMC23 | Olympiad | |
| pass@1 | - | 89.01 | 64.00 | 10.00 | 37.50 | 27.00 | 45.50 |
| pass@5 | - | 95.45 | 78.00 | 16.67 | 67.50 | 41.10 | 59.74 |
| Majority Vote | - | 91.66 | 68.20 | 10.00 | 47.50 | 31.01 | 49.67 |
| USC w/ Qwen 3B | 8.48 | 61.18 | 42.80 | 6.67 | 17.50 | 15.43 | 28.72 |
| USC w/ Qwen 7B | 5.89 | 5.89 | 61.20 | 6.67 | 47.50 | 28.78 | 43.83 |
| LLama ORM (8B) | 0.61 | 93.1 | 67.40 | **13.33** | 47.50 | 28.64 | 49.99 |
| Qwen PRM (7B) | 0.64 | 92.57 | 69.40 | **13.33** | **57.50** | 32.05 | 52.97 |
| Qwen PRM (72B) | 5.86 | 92.87 | 69.6 | **13.33** | **57.50** | 33.68 | 53.40 |
| Shepherd PRM (7B) | 0.65 | 90.75 | 64.40 | **13.33** | 35.00 | 27.00 | 46.10 |
| SSA RL (0.5B) | 0.20 | 92.65 | 75.40 | 10.00 | **57.50** | 37.98 | 54.71 |
| SSA RL (1.5B) | 0.33 | 92.49 | 76.60 | 10.00 | 52.50 | 38.72 | 54.06 |
| SSA RL (3B) | 0.55 | **93.25** | **76.80** | **13.33** | **57.50** | **39.76** | **56.13** |

finetune the model. We use learning rates 5e-5, 2e-5, and 1e-5 for 0.5B, 1.5B, and 3B correspondingly. For RL objective, we use GRPO method described in the Section 3.2. For SFT + RL objective, we use the trained model from the SFT objective, then we continue to train the RL objective on top of it. For each variant, we present models of three different sizes, trained from Qwen-0.5B, 1.5B, and 3B base models checkpoint respectively.

As an ablation for the SSA (RL), we also present SSA trained on GSM8K data only. Another ablation is to train the SSA RL version without thinking. It helps us to understand how each design decisions affect the output and performance under controlled settings.

## 5 MAIN RESULTS

### 5.1 COMPARISON WITH PARALLEL SCALING BASELINES

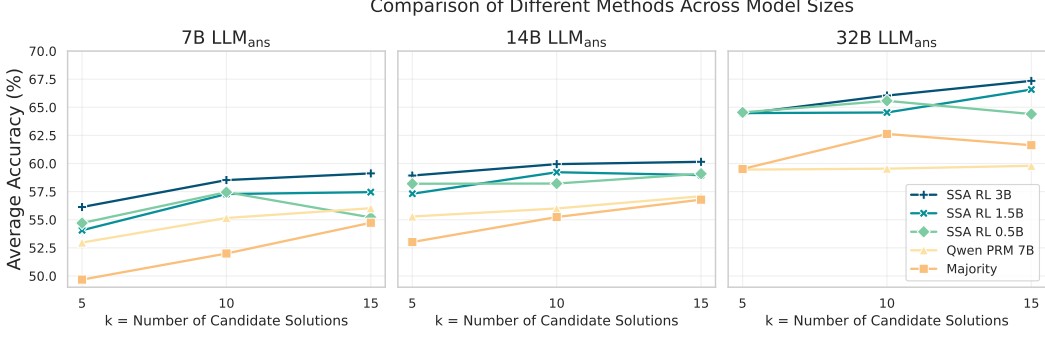

Figure 2: Compare the performance of SSA RL, PRM, and Majority Vote methods across Qwen 2.5 $LLM_{ans}$ model sizes (7B, 14B, 32B) and number of candidate solutions $k = 5, 10, 15$.

We report performances across GSM8K, MATH, AIME24, AMC23, Olympiad benchmark in Table 1, more detailed results can be seen in Table 4 Appendix A.

Overall, the results suggest that SSA is highly effective at leveraging multiple sampled solutions. In comparison to baseline methods, SSA substantially narrows the performance gap relative to the oracle (pass@5). Notably, even the smallest SSA variant leads to strong gains. SSA (0.5B) achieves an average performance of 54.7%, even outperforming the Qwen PRM (7B) baseline which uses a much larger model. We focus on SSA RL results in this section and leave more detailed results including SSA SFT and SSA SFT + RL in Appendix A.

Table 2: Generalization results for different model families (accuracy %) with $k = 5$ candidate from Llama 3.1 8B Instruct LLM$_{ans}$ models. Aggregation overhead (seconds) is measured end to end with AMC23 40 questions with k=5.

| Method | Aggregation Overhead (s) | Datasets | | | | | Avg |
|---|---|---|---|---|---|---|---|
| | | GSM8K | MATH | AIME24 | AMC23 | Olympiad | |
| Majority Vote | - | 87.95 | 50.40 | 6.67 | 35.00 | 17.66 | 39.54 |
| USC w/ Qwen 7B | 5.89 | 84.15 | 51.6 | 6.67 | 37.5 | 19.29 | 39.84 |
| LLama ORM (8B) | 0.61 | 89.61 | 51.8 | **13.33** | 30.00 | 18.4 | 40.63 |
| Qwen PRM (7B) | 0.64 | **91.51** | 56.20 | **13.33** | **35.00** | **20.77** | **43.36** |
| SSA RL (0.5B) | 0.20 | 88.17 | 52.80 | 10.00 | 30.00 | 20.18 | 40.23 |
| SSA RL (1.5B) | 0.33 | 88.48 | 56.60 | 10.00 | 27.50 | 20.47 | 40.61 |
| SSA RL (3B) | 0.55 | 89.08 | **57.80** | 10.00 | 32.50 | 20.62 | 42.00 |

## 5.2 GENERALIZATION CAPABILITIES OF SSA.

**Generalization across scales.** SSA trained on outputs from a smaller answer model (Qwen 2.5 7B Instruct) can generalize to outputs from larger answer models (14B, 32B). Figure 2 compare the overall performance of SSA and the other baselines under different numbers of samples. We can see that SSA RL consistently outperform the baselines under different number of candidate solution samples across Qwen models with scales ranging from 7B to 32B. More SSA versions and results are in Table 4 in the Appendix A

**Generalization across model families.** In addition to the Qwen 2.5 7B Instruct model, we also tested Llama 3.1 8B Instruct model for inference (Grattafiori et al., 2024).

Results are in Table 2. We can see the SSA method can also generalize well outside of the training data distribution and outperforming Majority Vote, USC, and ORM methods. While Qwen PRM performs slightly better than SSA RL 3B here (+1.36%), it requires substantially more training data (over 3 million examples , x21 times) and a larger model (7B).

**Generalization to harder datasets.** In Table 1, while we have trained the SSA models with problems only from GSM8K and MATH, we observe substantial performance gains on other harder datasets (AMC23 and Olympiad). Such results indicate SSA can **generalize to unseen test sets** outside of the training datasets.

In addition, we tested SSA on general tasks (specifically ARC-C, MMLU-Pro, and TruthfulQA (Clark et al., 2018; Wang et al., 2024c; Lin et al., 2022)) without additional training. We observe minor gains compared to majority vote, especially when using both SFT and RL, in ARC-C and MMLU-Pro dataset, but very little for TruthfulQA dataset. Further research is needed to study generalization across different domains. More details can be found in Appendix A.8.

## 5.3 COMPARISON WITH SEQUENTIAL TEST-TIME SCALING

We have demonstrated the advantageous of SSA over diverse parallel scaling methods in early §5.1. We now compare the performance on SSA and reasoning models that are enabled to spend more test-time compute sequentially with an RL-based approach as in (DeepSeek-AI et al., 2025). In particular, we use the results reported by Simple-RL Zoo from Zeng et al. (2025a) because they also uses Qwen-series models and train them on GSM8K and MATH dataset. In our comparison, we match the test-time computation, specifically, our 3B SSA is trained with Qwen-2.5-7B answers, but we apply it on top of 7B, 14B, 32B LLM$_{ans}$ models. We sample 8 candidate solutions from the LLM$_{ans}$ model

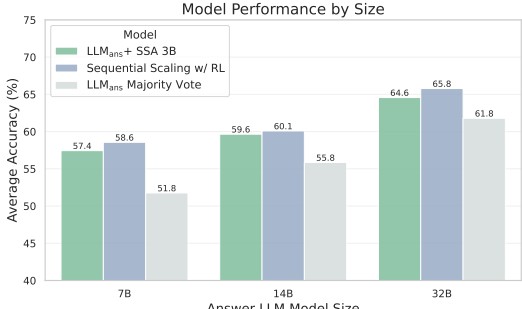

Figure 3: Performance comparison between sequential scaling with RL and our SSA

Table 3: Performance of SSA (accuracy, % averaged over 5 datasets) under increasing number of samples $K$. $*$ denote runs that use the adapted two-stage SSA (applied when $K>=32$). SSA at $K = 15, 64$ even outperform majority voting and PRM at $K = 128$.

| Answer K | 15 | 32 | 64 | 128 |
|---|---|---|---|---|
| Majority Vote | 54.73 | 54.07 | 54.43 | 55.11 |
| Qwen PRM (7B) | 56.02 | 55.37 | 58.49 | 56.58 |
| SSA RL (3B) | 59.12 | 58.42$^*$ | 59.78$^*$ | 58.68$^*$ |

with a maximum token length of 1,024 per answer, roughly matching the maximal generation length, 8,192, of these reasoning models.

As shown in Figure 3, both the SSA and the RL training from sequential scaling improve over majority voting. While SSA slightly underperforms sequential scaling with RL training, note that we only optimize the 3B model as opposed to the 7B, 14B, and 32B model.

### 5.4 SCALING SSA TO HANDLE A LARGER SAMPLE SET

In this section, we study SSA performance at larger $K$ values ($K$=32–128). So far, SSA considers all $K$ answers as a concatenated single input. However, concatenation at larger $K$ quickly runs into context length limitations, leading to performance degradation or failures in answer extraction. For instance, with $K = 32$, the concatenated input length approaches $\sim$30K tokens, nearly saturating the 32K context window of our backbone model (Qwen-2.5-3B). Furthermore, prior work highlights that the effective usable window is often shorter than the nominal size (Yang et al., 2025a; Liu et al., 2024; Hsieh et al., 2024; Ye et al., 2025).

**Two-Stage SSA** To address this, we introduce a simple yet effective two-stage adaptation of SSA for large $K$. In the first stage, we evenly split the $K$ samples into $l_2$ groups of size $l_1$, and run SSA independently on each group to produce $l_2$ intermediate candidates. In the second stage, SSA is applied to aggregate these $l_2$ candidates into a final answer. Instead of running SSA once, with this two-stage adaptation, we run SSA for $l_2 + 1$ times, $l_2$ times to aggregate $l_1$ inputs, and once to aggregate $l_2$ inputs. We include full descriptions of two stage SSA in Appendix A.5. For our experiments, we set the hyperparameter $l_1 = 15$. This sets $l_2 = 3$ for $K = 32$, and $l_2 = 5$ for $K = 64$ and $l_2 = 9$ for $K = 128$.

**Results** Table 3 presents results of SSA under increasing $K$ on Qwen-2.5-7B outputs, compared against majority voting and PRM-7B. We find that SSA makes more efficient use of samples, achieving strong performance. For example, SSA at $K = 15$ already surpasses both majority vote and PRM at $K = 128$. By contrast, PRM shows less stable performance (with noticeable fluctuations across $K$), as it operates by scoring individual answers without cross-sample aggregation, making it more vulnerable to the noisy reward model.

## 6 ANALYSES

### 6.1 COMPARING RL VS SFT

As an alternative to RL, we can train the SSA via supervised fine-tuning (SFT). How would SFT compare to RL? In our setup, we use the dataset described in Section 3.3 and fine-tune 0.5B, 1.5B, and 3B SSA models on this data for one epoch (learning rates 5e-5, 2e-5, and 1e-5, respectively). For comparison, we also train SSA models with *only* RL (on the same GSM8K problems for SFT) for controlled settings. Figure 4a and Table 6 (Appendix A.2) summarize the results.

We observe that SFT slightly outperforms RL in 0.5B, suggesting that direct supervision from high-quality data can help small model. For larger models (1.5B, 3B), RL yields better accuracy and robustness to larger $k$ during inference. SFT trained on LLM$_{ans}$ with $k = 5$ generalizes less effectively to $k = 10$. Overall, both SFT and RL can work well, but SFT's performance relies heavily on dataset quality and alignment. We also note that SFT tends to produce more readable reasoning traces, whereas RL outputs are often minimal (example outputs can be found in Appendix C).

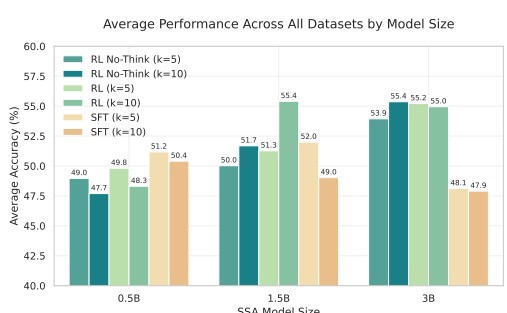
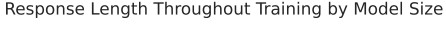
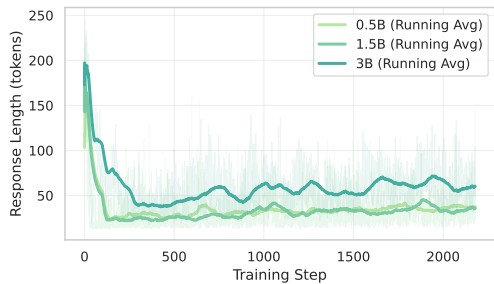

(a) Performance comparison of different training methods (SFT, No-Think, RL) across model sizes.

(b) Response length evolution during training for different model sizes.

Figure 4: Training method performance and response length analysis. (a) Average accuracy across datasets shows RL method is more generalizable than SFT method, with performance improving for larger models. (b) Response length trends during training show a rapid decrease of output length.

## 6.2 TO THINK OR NOT TO THINK

A distinct pattern we noted in our RL training is the reduction of the thinking tokens. Figure 4b shows how response length quickly drops during training across all model sizes. The model often simply repeats the provided instruction format for thinking (e.g., '<think>reasoning process here</think>') followed immediately with the final answer. Examples are shown in Appendix C Figure 14. This contrasts with other RL-based reasoning models that generate longer explanations with more training (DeepSeek-AI et al., 2025; Zeng et al., 2025a). It is likely because our SSA is conditioned on multiple candidate solutions, reducing the utility of detailed reasoning.

To assess the necessity of explicit reasoning tokens, we train an RL model variant without the reasoning step ('No Think'), where the model directly generates the answer (see Appendix B for prompt and Appendix C Figure 15 for example). Figure 4a summarizes the results, and Table 6 in Appendix A.2 compares this variant with the original approach across three model sizes in details. We observe minor performance degradation without explicit thinking, suggesting detailed reasoning tokens might not substantially contribute to the final performance in our current setup.

## 7 CONCLUSION

We introduce SSA, a small LM trained with RL that can leverage outputs from a larger base LLM. By decoupling RL training from base model, it suggests that the quality of the base model knowledge is more important for performance. This novel hybrid approach, blending parallel and sequential scaling methods, provides practical benefits for plug and play.

**Limitations** Across benchmarks, SSA succeeds mainly by picking an correct candidate among input candidate answers. Failures happens mainly when the gold answer is absent among the candidates (Appendix A.6 quantifies this trend). We experimented with enabling SSA to generate new final answers by cutting the last 10% of candidate answer tokens, but this does not yield better performance (See Appendix A.7 for details).

**Future work** We outline few possibilities for further improving and extending our approach. Promising directions include scaling the number of outputs to be aggregated, as well as building SSA that can incorporate outputs from multiple LLMs. Improving its performance for diverse application beyond mathematical reasoning, as well as enhacing SSA's new target answer synthesis ability can be fruitful. Overall, we believe SSA's hybrid approach offers a promising direction for future research in LLM reasoning.

## 8 REPRODUCIBILITY STATEMENT

Implementation details are describe in the Section 4. We also provide relevant code as supplementary materials.

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

# A  MORE RESULTS DETAILS

## A.1  SSA RESULTS

We compared SSA performance across model size, number of candidate solution $k$ size, subcategories, and different version of SSAs.

**SSA RL performance**  We can see SSA has strong performance gain in most of the sub categories compared to other methods in Figure 5. In addition, we also see consistent performance gain from 0.5B, 1B, and 3B model across different LLM$_{ans}$ sizes, and from Figure 6, we can see that LLM$_{ans}$ has more effect over the performance gain than the SSA model size.

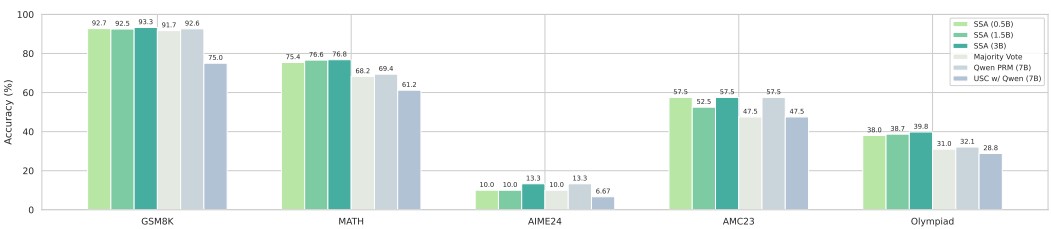

Figure 5: Compare the performance of model based on Qwen 2.5 7B with $k = 5$. SSAs are in green. We see SSA method is very effective against baseline methods.

In addition, we report all benchmark breakdon performance of SSA compared to the PRM 7B and majority vote in Table 4.

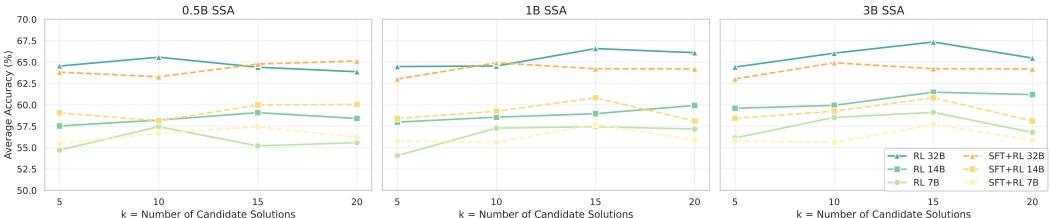

Figure 6: Compare each SSA's performance on the average of 5 benchmarks with both RL and SFT+RL methods. SFT+RL details are in Section A.3. The same SSAs are used on top of Qwen 2.5 7B, 14B, and 32B outputs. For each model, the parallel number of candidate solutions $k$ for SSAs are $k = 5, 10, 15, 20$.

We also report the statistical significance of all results in Table 5. For statistical testing, we employed McNemar's test, which is appropriate for paired binary outcomes. On our combined dataset analysis (aggregating all 2563 test examples across five benchmarks), our improvements over the baseline are statistically significant (p<0.05) for all configurations.

## A.2  ABLATION ON SFT, NO THINK, AND RL RESULTS

We put a combined table of SFT, No Think and RL in Table 6.

## A.3  COMBINING SFT AND RL

Because we observe that the model does not give human interpretable reasoning process, we wonder whether we can improve it. Understanding the model reasoning is important for the decision making process. So we decided to use the SFT version to cold start the model behavior. We hope to have a model to give reasoning process on why they select certain answers. We will call the cold started model as SSA SFT+RL. The performance comparison is in Figure 6 and Table 7. Overall, we observe 1-2% drop of the performance with SSA SFT+RL. However, it gives us a stronger sense on the model decision making process. In addition, SFT+RL version seems to have a better performance on datasets outside of the math domain as our discussion in Section A.8

Table 4: Performance (%) on five mathematical benchmarks for 7 B, 14 B and 32 B inference models trained with RL or RL + SFT.

| Benchmark | Method | 7B RL | | | | 7B RL+SFT | | | | 14B RL | | | | 14B RL+SFT | | | | 32B RL | | | | 32B RL+SFT | | | |
|---|---|---|---|---|---|---|---|---|---|---|---|---|---|---|---|---|---|---|---|---|---|---|---|---|---|
| | | 5 | 10 | 15 | 20 | 5 | 10 | 15 | 20 | 5 | 10 | 15 | 20 | 5 | 10 | 15 | 20 | 5 | 10 | 15 | 20 | 5 | 10 | 15 | 20 |
| GSM8K | Pass@k | 95.45 | 96.21 | 96.36 | 96.66 | 95.45 | 96.21 | 96.36 | 96.66 | 96.66 | 97.19 | 97.50 | 97.65 | 96.66 | 97.19 | 97.50 | 97.65 | 97.12 | 97.50 | 97.88 | 97.88 | 97.12 | 97.50 | 97.88 | 97.88 |
| | Majority vote | 91.66 | 91.96 | 92.72 | 92.65 | 91.66 | 91.96 | 92.72 | 92.65 | 94.62 | 94.54 | 94.31 | 94.39 | 94.62 | 94.54 | 94.31 | 94.39 | 95.30 | 95.75 | 95.98 | 95.98 | 95.30 | 95.75 | 95.98 | 95.98 |
| | Qwen PRM 7B | 92.57 | 93.18 | 93.40 | 93.33 | 92.57 | 93.18 | 93.40 | 93.33 | 95.38 | 95.91 | 96.13 | 96.13 | 95.38 | 95.91 | 96.13 | 96.13 | 96.13 | 96.59 | 96.74 | 96.82 | 96.13 | 96.59 | 96.74 | 96.82 |
| | Qwen PRM 72B | 92.87 | 93.71 | 93.86 | 94.09 | 92.87 | 93.71 | 93.86 | 94.09 | 95.68 | 96.06 | 96.44 | 96.29 | 95.68 | 96.06 | 96.44 | 96.29 | 95.75 | 96.06 | 96.44 | 96.13 | 95.75 | 96.06 | 96.44 | 96.13 |
| | 0.5B | 92.65 | 92.87 | 92.87 | 92.65 | 92.65 | 92.87 | 92.87 | 92.65 | 94.62 | 94.92 | 94.62 | 94.77 | 94.62 | 94.84 | 94.77 | 94.77 | 95.91 | 96.06 | 96.51 | 96.29 | 95.91 | 96.13 | 96.21 | 96.29 |
| | 1.5B | 92.49 | 92.72 | 93.25 | 93.10 | 92.49 | 92.72 | 93.25 | 93.10 | 94.39 | 94.54 | 95.07 | 94.92 | 94.84 | 94.62 | 94.77 | 94.77 | 96.13 | 96.29 | 96.21 | 96.29 | 95.98 | 95.98 | 96.21 | 96.21 |
| | 3B | 93.25 | 93.18 | 93.63 | 93.33 | 93.25 | 93.18 | 93.63 | 93.33 | 94.77 | 94.92 | 95.07 | 94.92 | 94.69 | 94.62 | 94.69 | 94.77 | 96.29 | 96.36 | 96.44 | 96.36 | 95.98 | 96.21 | 96.06 | 95.91 |
| MATH | Pass@k | 78.00 | 81.40 | 83.40 | 85.00 | 78.00 | 81.40 | 83.40 | 85.00 | 82.00 | 84.20 | 85.60 | 85.80 | 82.00 | 84.20 | 85.60 | 85.80 | 81.80 | 84.00 | 85.80 | 86.40 | 81.80 | 84.00 | 85.80 | 86.40 |
| | Majority vote | 68.20 | 69.40 | 71.60 | 71.20 | 68.20 | 69.40 | 71.60 | 71.20 | 74.40 | 75.20 | 75.00 | 75.40 | 74.40 | 75.20 | 75.00 | 75.40 | 75.00 | 75.40 | 75.40 | 75.00 | 75.00 | 75.40 | 75.40 | 75.00 |
| | Qwen PRM 7B | 69.40 | 69.00 | 69.20 | 70.00 | 69.40 | 69.00 | 69.20 | 70.00 | 73.60 | 74.40 | 74.20 | 75.20 | 73.60 | 74.40 | 74.20 | 75.20 | 73.20 | 74.00 | 74.00 | 74.20 | 73.20 | 74.00 | 74.00 | 74.20 |
| | Qwen PRM 72B | 69.6 | 70.4 | 71.4 | 72.2 | 69.6 | 70.4 | 71.4 | 72.2 | 72.8 | 74.6 | 75 | 75.2 | 72.8 | 74.6 | 75 | 75.2 | 72.8 | 74.6 | 75 | 74.6 | 72.8 | 74.6 | 75 | 74.6 |
| | 0.5B | 75.40 | 76.20 | 76.80 | 78.20 | 75.40 | 76.80 | 78.60 | 79.60 | 80.60 | 81.20 | 80.40 | 80.40 | 80.60 | 81.40 | 81.80 | 81.60 | 82.40 | 82.20 | 83.00 | 81.40 | 83.00 | 82.40 | 83.00 | 83.40 |
| | 1.5B | 76.60 | 77.60 | 78.60 | 79.40 | 76.80 | 77.60 | 78.00 | 78.40 | 81.40 | 81.20 | 81.60 | 82.40 | 81.80 | 81.80 | 81.60 | 81.00 | 83.60 | 84.00 | 83.80 | 84.00 | 83.60 | 83.80 | 83.80 | 83.00 |
| | 3B | 76.80 | 78.80 | 79.20 | 79.80 | 77.20 | 77.40 | 79.20 | 79.20 | 82.00 | 82.60 | 82.60 | 82.40 | 80.80 | 81.40 | 82.40 | 82.00 | 82.80 | 83.20 | 83.60 | 83.00 | 82.80 | 83.40 | 83.60 | 84.60 |
| AIME24 | Pass@k | 16.67 | 20.00 | 26.67 | 30.00 | 16.67 | 20.00 | 26.67 | 30.00 | 16.67 | 23.33 | 26.67 | 30.00 | 16.67 | 23.33 | 26.67 | 30.00 | 30.00 | 33.33 | 33.33 | 33.33 | 30.00 | 33.33 | 33.33 | 33.33 |
| | Majority vote | 10.00 | 13.33 | 16.67 | 16.67 | 10.00 | 13.33 | 16.67 | 16.67 | 10.00 | 13.33 | 20.00 | 16.67 | 10.00 | 13.33 | 20.00 | 16.67 | 20.00 | 30.00 | 26.67 | 30.00 | 20.00 | 30.00 | 26.67 | 30.00 |
| | Qwen PRM 7B | 13.33 | 16.67 | 20 | 23.33 | 13.33 | 16.67 | 20 | 23.33 | 13.33 | 13.33 | 13.33 | 13.33 | 13.33 | 13.33 | 13.33 | 13.33 | 20 | 20 | 23.33 | 23.33 | 20 | 20 | 23.33 | 23.33 |
| | Qwen PRM 72B | 13.33 | 16.67 | 6.67 | 13.33 | 13.33 | 16.67 | 13.33 | 13.33 | 13.33 | 16.67 | 16.67 | 13.33 | 13.33 | 16.67 | 16.67 | 13.33 | 20 | 30.00 | 26.67 | 30 | 20 | 23.33 | 23.33 | 30 |
| | 0.5B | 10.00 | 16.67 | 16.67 | 13.33 | 10.00 | 13.33 | 13.33 | 13.33 | 13.33 | 16.67 | 16.67 | 13.33 | 13.33 | 16.67 | 16.67 | 13.33 | 23.33 | 30.00 | 26.67 | 23.33 | 20.00 | 23.33 | 23.33 | 26.67 |
| | 1.5B | 10.00 | 20.00 | 16.67 | 16.67 | 10.00 | 10.00 | 16.67 | 16.67 | 13.33 | 20.00 | 20.00 | 13.33 | 16.67 | 20.00 | 20.00 | 16.67 | 23.33 | 26.67 | 30.00 | 26.67 | 23.33 | 30.00 | 23.33 | 26.67 |
| | 3B | 13.33 | 16.67 | 16.67 | 13.33 | 13.33 | 13.33 | 10.00 | 13.33 | 10.00 | 16.67 | 16.67 | 13.33 | 10.00 | 16.67 | 16.67 | 13.33 | 23.33 | 30.00 | 30.00 | 26.67 | 23.33 | 30.00 | 26.67 | 26.67 |
| AMC23 | Pass@k | 67.50 | 80.00 | 85.00 | 85.00 | 67.50 | 80.00 | 85.00 | 85.00 | 72.50 | 80.00 | 80.00 | 82.50 | 72.50 | 80.00 | 80.00 | 82.50 | 75.00 | 82.50 | 87.50 | 90.00 | 75.00 | 82.50 | 87.50 | 90.00 |
| | Majority vote | 47.50 | 52.50 | 57.50 | 57.50 | 47.50 | 52.50 | 57.50 | 57.50 | 50.00 | 55.00 | 55.00 | 60.00 | 50.00 | 55.00 | 55.00 | 60.00 | 67.50 | 70.00 | 67.50 | 65.00 | 67.50 | 70.00 | 67.50 | 65.00 |
| | Qwen PRM 7B | 57.50 | 62.50 | 62.50 | 60.00 | 57.50 | 62.50 | 62.50 | 70 | 60.00 | 60.00 | 65.00 | 65.00 | 60.00 | 60.00 | 65.00 | 65.00 | 65.00 | 67.50 | 65.00 | 62.50 | 65.00 | 67.50 | 65.00 | 62.50 |
| | Qwen PRM 72B | 57.5 | 70 | 70 | 70 | 57.5 | 70 | 62.5 | 70 | 62.5 | 57.5 | 60 | 60 | 62.5 | 57.5 | 60 | 60 | 57.5 | 62.5 | 60 | 65 | 57.5 | 62.5 | 60 | 65 |
| | 0.5B | 57.50 | 62.50 | 60.00 | 55.00 | 60.00 | 60.00 | 62.50 | 55.00 | 57.50 | 57.50 | 62.50 | 57.50 | 62.50 | 57.50 | 65.00 | 67.50 | 75.00 | 72.50 | 75.00 | 72.50 | 65.00 | 67.50 | 75.00 | 72.50 |
| | 1.5B | 52.50 | 55.00 | 57.50 | 55.00 | 60.00 | 57.50 | 60.00 | 52.50 | 55.00 | 57.50 | 55.00 | 62.50 | 57.50 | 57.50 | 65.00 | 60.00 | 65.00 | 67.50 | 75.00 | 75.00 | 70.00 | 67.50 | 70.00 | 67.50 |
| | 3B | 57.50 | 60.00 | 65.00 | 55.00 | 52.50 | 52.50 | 65.00 | 52.50 | 65.00 | 62.50 | 62.50 | 67.50 | 60.00 | 60.00 | 67.50 | 65.00 | 70.00 | 72.50 | 77.50 | 75.00 | 70.00 | 67.50 | 75.00 | 70.00 |
| Olympiad | Pass@k | 41.10 | 47.18 | 50.15 | 51.48 | 41.10 | 47.18 | 50.15 | 51.48 | 46.29 | 51.48 | 54.15 | 56.23 | 46.29 | 51.48 | 54.15 | 56.23 | 50.30 | 54.15 | 56.38 | 58.01 | 50.30 | 54.15 | 56.38 | 58.01 |
| | Majority vote | 31.01 | 32.79 | 35.16 | 35.31 | 31.01 | 32.79 | 35.16 | 35.31 | 36.05 | 38.13 | 39.61 | 39.76 | 36.05 | 38.13 | 39.61 | 39.76 | 39.76 | 41.99 | 42.58 | 42.88 | 39.76 | 41.99 | 42.58 | 42.88 |
| | Qwen PRM 7B | 32.05 | 34.42 | 35.01 | 33.83 | 32.05 | 34.42 | 35.01 | 33.83 | 34.12 | 36.35 | 36.80 | 36.94 | 34.12 | 36.35 | 36.80 | 36.94 | 39.61 | 39.61 | 39.91 | 40.50 | 39.61 | 39.61 | 39.91 | 40.50 |
| | Qwen PRM 72B | 33.68 | 35.46 | 35.91 | 35.31 | 33.68 | 35.46 | 35.91 | 35.31 | 35.76 | 38.58 | 39.76 | 40.21 | 35.76 | 38.58 | 39.76 | 40.21 | 40.65 | 40.95 | 41.25 | 41.54 | 40.65 | 40.95 | 41.25 | 41.54 |
| | 0.5B | 37.98 | 39.02 | 40.06 | 38.72 | 37.98 | 39.17 | 39.91 | 40.21 | 41.54 | 40.80 | 41.25 | 42.73 | 40.95 | 40.50 | 41.69 | 43.03 | 45.55 | 47.03 | 45.70 | 45.85 | 45.25 | 47.03 | 45.29 | 46.88 |
| | 1.5B | 38.72 | 41.10 | 41.25 | 41.69 | 39.47 | 39.91 | 40.80 | 39.76 | 42.43 | 42.88 | 43.47 | 43.18 | 41.69 | 42.14 | 42.73 | 41.84 | 46.88 | 48.22 | 47.92 | 48.52 | 47.03 | 47.33 | 47.77 | 47.63 |
| | 3B | 39.76 | 40.65 | 41.10 | 42.43 | 38.43 | 40.36 | 40.95 | 41.54 | 42.88 | 43.03 | 43.92 | 44.51 | 41.99 | 43.03 | 43.77 | 43.92 | 47.18 | 48.22 | 49.41 | 48.81 | 46.44 | 46.88 | 47.33 | 48.52 |
| Average | Pass@k | 59.74 | 64.96 | 68.32 | 69.63 | 59.74 | 64.96 | 68.32 | 69.63 | 62.82 | 67.24 | 68.78 | 70.44 | 62.82 | 67.24 | 68.78 | 70.44 | 66.84 | 70.30 | 72.18 | 73.12 | 66.84 | 70.30 | 72.18 | 73.12 |
| | Majority vote | 49.67 | 52.00 | 54.73 | 54.67 | 49.67 | 52.00 | 54.73 | 54.67 | 53.01 | 55.24 | 56.78 | 57.24 | 53.01 | 55.24 | 56.78 | 57.24 | 59.51 | 62.63 | 62.58 | 61.77 | 59.51 | 62.63 | 62.58 | 61.77 |
| | Qwen PRM 7B | 52.97 | 55.15 | 56.02 | 56.10 | 52.97 | 55.15 | 56.02 | 56.10 | 55.29 | 56.00 | 57.09 | 57.32 | 55.29 | 56.00 | 57.09 | 57.32 | 59.45 | 59.54 | 59.80 | 59.47 | 59.45 | 59.54 | 59.80 | 59.47 |
| | Qwen PRM 72B | 53.40 | 55.75 | 58.23 | 58.99 | 53.40 | 55.75 | 58.23 | 58.99 | 56.01 | 56.01 | 56.91 | 57.01 | 56.01 | 56.01 | 56.91 | 57.01 | 57.34 | 58.82 | 59.87 | 61.45 | 57.34 | 58.82 | 59.87 | 61.45 |
| | 0.5B | 54.71 | 57.45 | 57.45 | 55.58 | 55.47 | **56.67** | 57.43 | 56.22 | **59.07** | 58.22 | 59.09 | 57.75 | **59.07** | 58.14 | 59.99 | **60.05** | **63.83** | 63.28 | 64.77 | 63.87 | **63.83** | 63.28 | 64.77 | 63.87 |
| | 1.5B | 54.06 | 57.28 | 57.45 | 57.17 | **55.77** | 55.61 | 57.71 | 55.87 | 58.41 | 59.22 | 58.97 | 59.27 | 58.41 | **59.26** | 60.82 | 58.11 | 64.48 | 64.54 | 64.22 | **66.10** | 64.48 | 64.54 | **64.92** | **66.10** |
| | 3B | **56.13** | **58.53** | **59.12** | 56.78 | 54.82 | 55.31 | 57.68 | 55.90 | **58.93** | **59.94** | **60.15** | **60.53** | 57.50 | 59.14 | **61.01** | 59.80 | 64.42 | **66.04** | **67.35** | 65.47 | 63.71 | 64.80 | **65.73** | **65.14** |

Table 5: Statistical comparison of SSA models *vs.* baselines on mathematical benchmarks.

| Baseline | Method | 7B | | | | | | | | | 14B | | | | | | | | | 32B | | | | | | | | |
|---|---|---|---|---|---|---|---|---|---|---|---|---|---|---|---|---|---|---|---|---|---|---|---|---|---|---|---|---|
| | | 5 | | 10 | | 20 | | | | | 5 | | 10 | | 20 | | | | | 5 | | 10 | | 20 | | | | |
| | | p-val | sig | p-val | sig | p-val | sig | | | | p-val | sig | p-val | sig | p-val | sig | | | | p-val | sig | p-val | sig | p-val | sig | | | |
| **Majority Vote** | 0.5B vs. Maj | 1.1e-14 | * | 3.5e-14 | * | 1.9e-05 | * | | | | 7.0e-11 | * | 1.4e-06 | * | 4.5e-06 | * | | | | 2.5e-16 | * | 3.6e-10 | * | 2.5e-06 | * | | | |
| | 1B vs. Maj | 4.2e-17 | * | 2.0e-18 | * | 9.6e-14 | * | | | | 1.2e-12 | * | 1.3e-09 | * | 7.4e-12 | * | | | | 1.4e-18 | * | 1.8e-17 | * | 1.6e-16 | * | | | |
| | 3B vs. Maj | 5.8e-25 | * | 6.3e-23 | * | 5.1e-15 | * | | | | 3.1e-17 | * | 5.1e-14 | * | 4.3e-14 | * | | | | 2.0e-20 | * | 3.0e-15 | * | 8.4e-15 | * | | | |
| **Qwen PRM 7B** | 0.5B vs. PRM | 6.7e-06 | * | 1.0e-04 | * | 4.6e-04 | * | | | | 1.1e-08 | * | 1.8e-04 | * | 1.4e-03 | * | | | | 3.3e-13 | * | 2.8e-11 | * | 6.6e-07 | * | | | |
| | 1B vs. PRM | 1.1e-06 | * | 1.2e-06 | * | 2.7e-08 | * | | | | 9.3e-10 | * | 8.1e-06 | * | 9.0e-06 | * | | | | 1.0e-13 | * | 1.5e-15 | * | 2.7e-13 | * | | | |
| | 3B vs. PRM | 1.0e-10 | * | 1.4e-08 | * | 4.2e-10 | * | | | | 1.5e-13 | * | 2.0e-08 | * | 6.9e-08 | * | | | | 5.5e-15 | * | 3.5e-15 | * | 1.7e-12 | * | | | |

*Note*: All results are on the combined of all 5 benchmarks. The * indicates statistical significance ($p < 0.05$). All SSA models (0.5B, 1B, 3B) show significant improvement over both Majority vote and PRM baselines across all inference model sizes (7B, 14B, 32B).

Table 6: Ablation of different training methods. Including SFT method, No Thinking Method, and RL methods trained with LLM$_{ans}$ $k = 5$ on GSM8K train data only. We report LLM$_{ans}$ $k = 5, 10$ results as accuracy (%)

| Model | GSM8K k=5 | GSM8K k=10 | MATH k=5 | MATH k=10 | AIME24 k=5 | AIME24 k=10 | AMC23 k=5 | AMC23 k=10 | Olympiad k=5 | Olympiad k=10 | Average k=5 | Average k=10 |
|---|---|---|---|---|---|---|---|---|---|---|---|---|
| *Baseline* | | | | | | | | | | | | |
| Majority Vote | 91.66 | 91.96 | 68.20 | 69.40 | 10 | 13.33 | 47.50 | 52.50 | 31.01 | 32.79 | 49.67 | 52.0 |
| PRM | 92.57 | 93.18 | 69.4 | 69.00 | 13.33 | 16.67 | 57.5 | 62.5 | 32.05 | 34.42 | 52.97 | 55.15 |
| *0.5B* | | | | | | | | | | | | |
| SFT | 91.51 | 92.04 | 66.0 | 66.6 | 10.0 | 10.0 | 57.5 | 50.0 | 30.86 | 33.38 | 51.17 | 50.4 |
| RL No-Think | 91.43 | 91.58 | 62.4 | 61.2 | 10.0 | 10.0 | 52.5 | 47.5 | 28.49 | 28.19 | 48.96 | 47.69 |
| RL | 92.42 | 93.1 | 58.6 | 57.2 | 10.0 | 13.33 | 60.0 | 50.0 | 28.04 | 27.89 | 49.81 | 48.3 |
| *1.5B* | | | | | | | | | | | | |
| SFT | 91.51 | 91.66 | 72.4 | 70.4 | 10.0 | 10.0 | 50.0 | 42.5 | 36.05 | 30.56 | 51.99 | 49.02 |
| RL No-Think | 92.65 | 93.1 | 63.2 | 63.2 | 10.0 | 16.67 | 52.5 | 52.5 | 31.75 | 32.94 | 50.02 | 51.68 |
| RL | 92.65 | 93.1 | 71.6 | 73.0 | 10.0 | 20.0 | 47.5 | 55.0 | 34.57 | 35.91 | 51.26 | 55.4 |
| *3B* | | | | | | | | | | | | |
| SFT | 91.13 | 91.58 | 68.4 | 66.6 | 6.67 | 10.0 | 45.0 | 40.0 | 29.38 | 31.31 | 48.12 | 47.9 |
| RL No-Think | 92.8 | 93.25 | 72.6 | 72.8 | 10.0 | 20.0 | 57.5 | 52.5 | 36.8 | 38.28 | 53.94 | 55.37 |
| RL | 93.18 | 93.1 | 75.0 | 74.8 | 13.33 | 16.67 | 60.0 | 52.5 | 34.57 | 37.69 | 55.22 | 54.95 |

Table 7: Average Performance (%) of SSA RL and SSA SFT+RL over five benchmarks. The answers are generated with Qwen 32B models. The second row indicates the number of sampled answers (k).

| Method | RL 5 | RL 10 | RL 15 | RL 20 | SFT + RL 5 | SFT + RL 10 | SFT + RL 15 | SFT + RL 20 |
|---|---|---|---|---|---|---|---|---|
| Pass@k | 66.84 | 70.30 | 72.18 | 73.12 | 66.84 | 70.30 | 72.18 | 73.12 |
| Majority vote | 59.51 | 62.63 | 61.63 | 61.77 | 59.51 | 62.63 | 61.63 | 61.77 |
| Qwen PRM | 59.45 | 59.54 | 59.80 | 59.47 | 59.45 | 59.54 | 59.80 | 59.47 |
| SSA (0.5B) | **64.54** | 65.57 | 64.40 | 63.87 | **63.83** | 63.28 | 64.77 | 65.13 |
| SSA (1.5B) | 64.48 | 64.54 | 66.58 | **66.10** | 63.02 | **64.92** | 64.22 | 64.20 |
| SSA (3B) | 64.42 | **66.04** | **67.35** | 65.47 | 63.71 | 64.80 | **65.73** | **65.14** |

## A.4 INCREASING $k$ DURING TRAINING

We see that the inference $k$ could lead to the potential improvements of the performance. Would the same hold if we improve the $k$ during training. For the original design, we use $k = 5$ for the training. For comparison, we train the model with $k = 8$ to see its performance. The results are presented in Figure 7. We see that training longer context does not help with the performance. In fact it has lower performance on average for the dataset. It might due to longer context creates more same answers, and it would make the model to choose more depends on the majority vote than distinguish the differences.

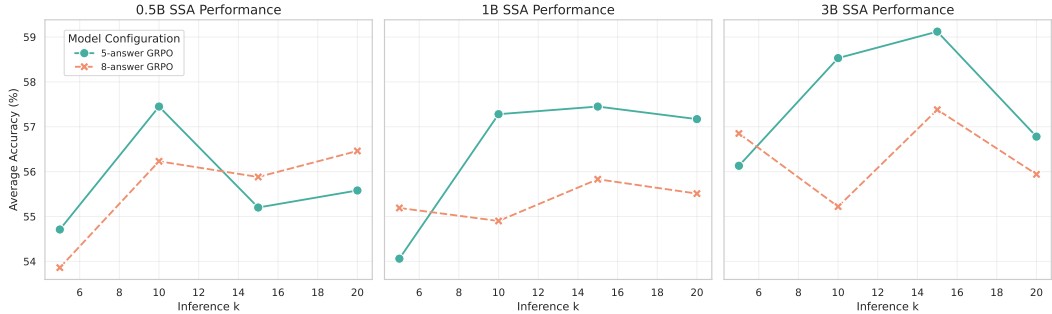

Figure 7: Compare training k and their effect. It seems that increasing context length to longer k during training does not out perform the shorter context.

Table 8: Aggregation overhead per question

| Answer K | 32 | 64 | 128 |
|---|---|---|---|
| Qwen PRM (7B) | 5.25 | 5.41 | 5.80 |
| SSA RL (3B) | 9.28 | 7.94 | 7.02 |

## A.5 SCALING K DURING INFERENCE AND TWO-STAGE SSA

To handle large $K$ under context limits, we use a simple two-stage adaptation of SSA.

**Stage 1.** We form $\ell_2$ overlapping groups by taking *evenly spaced, cyclic windows* of length $\ell_1$ over the $K$ candidates. Concretely, with start indices $s_i = \left\lfloor \frac{iK}{\ell_2} \right\rfloor$ for $i = 0, \ldots, \ell_2 - 1$, the $i$-th group is $G_i = \left[ a_{(s_i+t) \bmod K} \right]_{t=0}^{\ell_1-1}$. We run SSA independently on each $G_i$ to produce one intermediate winner.

**Stage 2.** We concatenate the $\ell_2$ intermediate winners and run SSA once more to produce the final answer.

This makes $\ell_2+1$ SSA calls: $\ell_2$ calls on inputs of size $\ell_1$, then one call on $\ell_2$. We use $\ell_1=15$ in all experiments; thus $\ell_2=\lceil K/\ell_1 \rceil$ gives $\ell_2=3$ for $K=32$, $\ell_2=5$ for $K=64$, and $\ell_2=9$ for $K=128$. By construction, groups *overlap* whenever $\ell_1 > K/\ell_2$ (our default), yielding a replication factor $(\ell_1\ell_2)/K$ that adds controlled redundancy for robustness at modest extra compute. The details are in Algorithm 1.

In addition to the performance reported in the Table 3, we also report the aggregation overhead over the AMC23 dataset. The results are in Table 8.

---

**Algorithm 1** Two-stage SSA

**Require:** Question $x$; $K$ candidates $A = \{a_0, \ldots, a_{K-1}\}$; group size $\ell_1$; group count $\ell_2$
1: **if** $K \leq \ell_1$ **then**
2:     **return** SSA$(x, A)$
3: **end if**
4: `stage1` $\leftarrow []$                                $\triangleright$ will hold $\ell_2$ winners (full CoTs)
5: **for** $i = 0$ to $\ell_2 - 1$ **do**             $\triangleright$ **Stage 1:** $\ell_2$ calls, each on $\ell_1$ answers
6:     $s_i \leftarrow \left\lfloor \dfrac{i \cdot K}{\ell_2} \right\rfloor$                         evenly spaced starts
7:     $G_i \leftarrow \left[ a_{((s_i+t) \bmod K)} : t = 0, \ldots, \ell_1 - 1 \right]$     cyclic window (overlap-allowed)
8:     $\hat{y}_i \leftarrow$ SSA$(x, G_i)$                   infer the group's final answer
9:     $a_{G_i} \leftarrow$ MapBack$(\hat{y}_i, G_i)$         pick the chosen candidate's *full* CoT
10:    `stage1`.append$(a_{G_i})$
11: **end for**
12: **return** SSA$(x, $ `stage1`$)$          $\triangleright$ **Stage 2:** aggregate $\ell_2$ winners

---

## A.6 ERROR ANALYSIS

We conduct error analysis of the SSA outputs. We use SSA 3B model with five answer samples. Across all datasets the vast majority of correct cases are simply *copied* from a candidate that already contains the ground-truth answer. Hence the main value of SSA lies in finding the right candidate. When the ground-truth answer is absent from the sample set, SSA fails in two ways: it either chooses the majority wrong answer ($y^* \notin C$ *copied*) or try to come up a new but still wrong answer ($y^* \notin C$ *synthesized*). While SSA sometimes come up with the correct answer even the ground truth are not provided in the inference, when we manually examine the answers in this category most are in wrong format that extraction failed. It confirms an input quality bottleneck: if no correct candidate exists, the policy has difficulty to recover it. Conversely, errors with a correct candidate present ($y^* \in C$ / *copied wrong* or $y^* \in C$ / *synthesized wrong*) are much smaller, showing that SSA rarely mis-ranks truly correct answers. In order to understand whether SSA can synthesize new answers, we designed addition experiments to cut off last 10% of the answers, and our results are in Appendix A.7. It

Table 9: Accuracy and error composition for the 3 B SSA ($k{=}5$). Columns are percentages of the whole dataset. $y^* \in C$ means ground-truth answer $y^*$ included among the 5 candidates $C$.

| | Acc.↑ | **Correct** | | **Wrong** | | | |
|---|---|---|---|---|---|---|---|
| | | $y^* \in C$ copied | $y^* \notin C$ synthesized | $y^* \notin C$ copied | $y^* \notin C$ synthesized | $y^* \in C$ copied | $y^* \in C$ synthesized |
| GSM8K | 93.3 | 87.7 | 5.5 | 3.3 | 1.0 | 2.3 | 0.2 |
| MATH | 79.2 | 75.6 | 3.6 | 9.4 | 3.6 | 6.6 | 1.2 |
| AMC23 | 57.5 | 52.5 | 5.0 | 12.5 | 7.5 | 12.5 | 10.0 |
| AIME24 | 13.3 | 13.3 | 0.0 | 23.3 | 50.0 | 3.3 | 10.0 |
| Olympiad | 39.8 | 34.9 | 4.9 | 27.6 | 24.6 | 5.6 | 2.4 |

shows that the SSA has synthesize ability when answers are all removed, and the accuracy has some degradation.

## A.7  COPY OR SYNTHESIZE?

Our error analysis reveals that the majority of correct answers are copied from the provided responses. To test whether the SSA can synthesize answers, we created a variant of the dataset where the last 10% of every answer is cut off. This ensures that the final answers are not directly available in the input, requiring the model to synthesize them.

Table 10: Performance comparison between original responses and those with last 10% removed.

| Method | **Datasets** | | | | | Avg |
|---|---|---|---|---|---|---|
| | GSM8K | MATH | AIME24 | AMC23 | Olympiad | |
| *Original (complete answers)* | | | | | | |
| SSA | 93.25 | 76.80 | 13.33 | 57.50 | 39.76 | **56.13** |
| *Truncated (last 10% removed)* | | | | | | |
| SSA + RL | 74.22 | 52.80 | 0.00 | 22.50 | 23.00 | **34.50** |
| SSA + RL + SFT | 89.99 | 67.80 | 6.67 | 40.00 | 31.01 | **47.09** |

As shown in Table 10, performance significantly drops when answers must be synthesized rather than copied. The RL method alone achieves only 34.50% average accuracy, while combining RL with SFT improves this to 47.09%. Notably, the RL+SFT approach recovers much of the performance gap, particularly on GSM8K (89.99% vs. original 93.25%), suggesting effective synthesis capabilities for easy problems. However, since pure copy would only result 0%, this suggests that SSA is not just copy but able to synethize correct results.

## A.8  GENERALIZATION TO OTHER TASKS

Table 11 reports accuracy on three general tasks benchmarks. On **ARC-C** and **MMLU-Pro** the SFT + RL SSA mostly beats the majority-vote baseline (Clark et al., 2018; Wang et al., 2024c). Pure RL SSAs give smaller (sometimes negative) gains, suggesting that an SFT warm-start improves out-of-domain transfer. Pure SFT SSAs exhibit varied performance: strong at 3B scale but significantly weaker at smaller scales (0.5B and 1.5B), suggesting SFT's generalization capability strongly depends on model capacity for out domain generalization. On **TruthfulQA** (truthfulness/adversarial) none of the SSAs can consistently outperform majority voting, suggesting that truthfulness might not be beneficial from answer selection. Tackling truthfulness might require different goal than reasoning among answers.

Table 11: Model evaluation results on ARC, MMLU-PRO, TruthfulQA benchmarks

| Metric | Qwen2.5-7B-Instruct | | SSA RL | | | SSA SFT + RL | | | SSA SFT | | |
|---|---|---|---|---|---|---|---|---|---|---|---|
| | pass@1 | Majority Vote (5) | 0.5B | 1.5B | 3B | 0.5B | 1.5B | 3B | 0.5B | 1.5B | 3B |
| ARC-C | 88.65 | 91.38 | 88.05 | 91.3 | 88.99 | 91.72 | 91.38 | 91.81 | 69.88 | 90.02 | **92.06** |
| MMLU-PRO | 43.63 | 49.24 | 33.19 | 43.09 | 39.84 | 46.28 | 48.01 | 50.37 | 25.76 | 43.93 | **51.41** |
| TruthfulQA | 62.43 | 66.23 | 66.52 | **67.11** | 64.62 | 66.08 | 64.47 | 63.89 | 49.85 | 63.89 | 66.23 |

Table 12: Efficiency Evaluation. We report end-to-end runtime on one RTX 6000 Ada.

| Method | Base-LM passes | Aggregator time | Total time (s) ↓ | Overall Accuracy (%) ↑ |
|---|---|---|---|---|
| Qwen2.5-7B | 1 | – | 556.12 | 45.5 |
| Majority vote ($k$=5) | 5×Qwen2.5-7B | – | **2 780.6** | 49.67 |
| Qwen PRM 7B ($k$=5) | 5×Qwen2.5-7B | 21.9 | 2 780.6 + 21.9 = **2 802.5** | 52.97 |
| SSA RL 3B ($k$=5, ours) | 5×Qwen2.5-7B | 25.7 | 2 780.6 + 25.7 = **2 806.3** | 56.13 |
| SimpleRL 7B (seq. RL) | 1 pass | – | **3 273.2** | 58.56 |

"Base-LM passes" = number of forward decodes of Qwen-2.5-7B-Instruct (556.12 s per pass). Aggregator time is measured separately.

## A.9 EFFICIENCY EVALUATION

We now evaluate the amount of compute required for each aggregation strategy. Compared to other methods which considers each sample independently, our method considers them jointly, generating longer sequence which can incur high computation cost.

**Compute Cost (Wall-clock)** For more details, we conduct an actual measure based on the real settings. The results are in Table 12. All measurements were taken on a single NVIDIA RTX 6000 Ada with the AMC23 benchmark (40 questions). In practice, parallelized sampling from the LLM can significantly speed up the time required.

**Inference Cost** To measure raw compute (flops), we follow the formula from Kaplan et al. (2020). It uses roughly $C_{forward} \approx 2N$, and the backward pass is about 2 times which is $4N$. So the total is $C \approx 6N$ FLOPs per training token, with $N$ as the model parameter and $C$ as the non embedding training compute. We use $D$ as token length.

For inference cost we will use $C_{forward} \approx 2ND$ per query. For k = 5, input context $D \approx 5 * 1000$. So the SSA method needs a 7B base model with inference cost $2ND = 2 * 7 * 10^9 * (1000 * 5) = 70 TFLOPs$. SSA itself has a constant compute overhead of $2ND = 2 * 3 * 10^9 * (5000 + 60) \approx 30 TFLOPs$ since SSA's compute cost depends on the input token length not the inference model size. The total for SSA would be $70 + 30 = 100 TFLOPs$. In comparison, the sequential RL approach would result $2ND = 2 * 7 * 10^9 * 8000 \approx 110 TFLOPs$. This method scales favourably to larger bases (SSA 350 TFLOPs vs Sequential RL 512 TFLOPs on a 32B model).

## A.10 STABILITY ANALYSES

### A.10.1 EFFECT OF CANDIDATE ORDERING

A potential concern for methods that concatenate $K$ candidates is that the relative order of those candidates might influence the SSA's decision. To test this, we evaluate the 3B SSA (RL) under three independent random permutations of the $K$ responses of 7B answer model at inference time, holding everything else fixed. As shown in Table 13, we do not observe meaningful differences when changing the order.

### A.10.2 TRAINING DIFFERENT RANDOM SEED

Reinforcement learning can exhibit sensitivity to initialization. We therefore train SSA 3B (RL) with another seed 16 using identical data and hyperparameters, and evaluate the resulting checkpoints under the same protocol as the main results of 7B answer model. Table 14 reports per-$K$ accuracy. We do not see significant differences.

## A.11 TRAINING DATA STATISTICS

Table 15 summarizes the distribution of instance-level correctness in the raw training pool prior to filtering. For each question we sampled $K$=5 candidate solutions using top-$k$ sampling. "$m/5$" indicates that exactly $m$ of the five candidates match the reference answer.

**Filtering and preprocessing.** We apply two light filters before RL training:

Table 13: **Ordering sensitivity.** SSA 3B (RL) evaluated with the original concatenation order vs. three random permutations at inference time. Values are accuracy (%).

| Metric | 5 | 10 | 15 | 20 |
|---|---|---|---|---|
| Original (%) | 56.13 | 58.53 | 59.12 | 56.78 |
| Avg. over 3 random permutations (%) | 56.06 | 58.70 | 59.40 | 58.47 |

Table 14: **Training-seed sensitivity.** SSA 3B (RL) trained with different random seeds. Values are accuracy (%).

| Metric | 5 | 10 | 15 | 20 |
|---|---|---|---|---|
| Original Random Seed 42(%) | 56.13 | 58.53 | 59.12 | 56.78 |
| Random Seed 16 (%) | 56.09 | 57.27 | 59.49 | 56.64 |

Table 15: **Correctness distribution** in the raw training pool (five sampled responses per question). Counts and column percentages are shown.

| Correct | GSM8K | MATH | Combined |
|---|---|---|---|
| 0/5 | 618 (8.3%) | 3170 (26.4%) | 3788 (19.5%) |
| 1/5 | 218 (2.9%) | 1007 (8.4%) | 1225 (6.3%) |
| 2/5 | 198 (2.6%) | 841 (7.0%) | 1039 (5.3%) |
| 3/5 | 268 (3.6%) | 881 (7.3%) | 1149 (5.9%) |
| 4/5 | 477 (6.4%) | 1157 (9.6%) | 1634 (8.4%) |
| 5/5 | 5694 (76.2%) | 4944 (41.2%) | 10638 (54.6%) |
| **Total** | **7473 (100.0%)** | **12000 (100.0%)** | **19473 (100.0%)** |

- **Validity filter.** We discard instances where more than one of the five samples is NULL (unparsable/empty). This removes $\approx 0.4\%$ of GSM8K and $\approx 9.8\%$ of MATH instances.

- **Length filter.** Answer sets exhibit a long-tailed length distribution; some concatenations exceed 8k tokens due to looping or unbounded reasoning. To control VRAM and remove pathological traces, we drop instances whose concatenated prompt + answers exceed 4k tokens. This reduces the pool from $\sim$19k to $\sim$17k instances and slightly denoises the supervision.

These filters are minimal (no step-level labeling) and aimed purely at stabilizing training; we did not tune them for accuracy.

# B PROMPT DETAILS

For SSA method, we trained and evaluate it with the following prompt:

---

**SSA Prompt:** A conversation between User and Assistant. The user provide a question and some proposed answers. The Assistant first evaluate each answers individually,check whether each answer directly addresses the original question, assess the correctness of each answer based on logical reasoning, calculations, and accuracy relative to the question. After thorough evaluation, identify one correct answer. If the correct answer is not in the provided proposed answers, the Assistant will combine the correct answer with the proposed answers and provide the correct answer. The reasoning process and answer are enclosed within <think></think> and <answer></answer> tags, respectively, i.e., <think>reasoning process here</think> <answer>answer here</answer>.

---

Figure 8: Example prompt for SSA. For reward extraction, we will use rule based extraction to extract anything inside <think></think> and <answer></answer>. If the output matches the structure and able to extract some values we will provide minimal format reward.

---

**SSA No-Think Prompt:**A conversation between User and Assistant. The user provide a question and some proposed answers. The Assistant answer the question based on the proposed answers. The answer is enclosed within <answer></answer> tag, i.e., <answer>answer here</answer>.

---

Figure 9: Example prompt for SSA

---

**USC Prompt:** You are a helpful assistant. The user provide a question and some proposed answers. The Assistant first evaluate each answers individually,check whether each answer directly addresses the original question, assess the correctness of each answer based on logical reasoning, calculations, and accuracy relative to the question. After thorough evaluation, identify one correct answer based on majority consensus. The reasoning process and answer are enclosed within <think></think> and <answer></answer> tags, respectively, i.e., <think>reasoning process here</think> <answer>answer here</answer>.

---

Figure 10: Example prompt for USC prompt to the Instruct model

---

**SFT Data Construction Prompt:** Here is a question and some proposed answers. You need to evaluate each answers individually, check whether each answer directly addresses the original question, assess the correctness of each answer based on logical reasoning, calculations, and accuracy relative to the question. After thorough evaluation, identify one correct answer. If the correct answer is not in the provided proposed answers, the Assistant will combine the correct partial responses to proposed answers and provide the correct answer. Make the reasoning process concise and to the point. The reasoning process and answer are enclosed within <think></think> and <answer></answer> tags, respectively,i.e., <think>reasoning process here</think> <answer>answer here</answer>. Here is the Question and proposed answers: **Question and 5 Answers**.
Here is the labeled answer with answer tags '<answer>**Ground Truth Answer**</answer>', and most likely the correct answer. If none of the proposed answers is correct, come up with the reasoning process to arrive at the actual correct final answer. **Attention: Do not ever mention the answer is given in the response, but you need to justify it from the problem and proposed answers!**

---

Figure 11: Example prompt for constructing SFT dataset

## C  EXAMPLE RESULTS

A conversation between User and Assistant. The user provide a question and some proposed answers. The Assistant first evaluate each answers individually,check whether each answer directly addresses the original question, assess the correctness of each answer based on logical reasoning, calculations, and accuracy relative to the question. After thorough evaluation, identify one correct answer. If the correct answer is not in the provided proposed answers, the Assistant will combine the correct answer with the proposed answers and provide the correct answer.The reasoning process and answer are enclosed within <think></think> and <answer></answer> tags, respectively, i.e., <think>reasoning process here</think> <answer>answer here</answer>.

User: Question: What is the value of\n\\[2^3 - 1^3 + 4^3 - 3^3 + 6^3 - 5^3 + \\dots + 18^3 - 17^3?\\]\n\nBelow are the proposed answers:\n\n

Answer 1:\nTo solve the given expression \\(2^3 - 1^3 + 4^3 - 3^3 + 6^3 - 5^3 + \\dots + 18^3 - 17^3\\), we can observe that it consists of pairs of terms where each pair is of the form \\( (2k)^3 - (2k-1)^3 \\).\n\nLet's simplify the general term in the series:\n\\[\n(2k)^3 - (2k-1)^3\\]\n\nUsing the binomial expansion for cubes, we have:\n\\[\na^3 - b^3 = (a - b)(a^2 + ab + b^2)\n\\]\nwhere \\(a = 2k\\) and \\(b = 2k-1\\). Substituting these values, we get:\n\\[\n(2k)^3 - (2k-1)^3 = [(2k) - (2k-1)][(2k)^2 + (2k)(2k-1) + (2k-1)^2]\n\\]\nSimplifying inside the brackets:\n\\[\n(2k - (2k-1)) = 1\n\\]\nSo,\n\\[\n(2k)^3 - (2k-1)^3 = 1 \\cdot [(2k)^2 + (2k)(2k-1) + (2k-1)^2]\n\\]\nNow, let's expand the quadratic expression:\n\\[\n(2k)^2 = 4k^2\n\\]\n\\[\n(2k)(2k-1) = 4k^2 - 2k\n\\]\n\\[\n(2k-1)^2 = 4k^2 - 4k + 1\n\\]\nAdding these together:\n\\[\n4k^2 + (4k^2 - 2k) + (4k^2 - 4k + 1) = 12k^2 - 6k + 1\n\\]\nThus,\n\\[\n(2k)^3 - (2k-1)^3 = 12k^2 - 6k + 1\n\\]\nNext, we need to sum this expression from \\(k=1\\) to \\(k=9\\) (since the sequence goes up to 18, which corresponds to \\(k=9\\)):\n\\[\n\\sum_{k=1}^{9} (12k^2 - 6k + 1)\n\\]\nWe can split this into three separate sums:\n\\[\n\\sum_{k=1}^{9} 12k^2 - \\sum_{k=1}^{9} 6k + \\sum_{k=1}^{9} 1\n\\]\nCalculating each sum separately:\n1. Sum of squares:\n\\[\n\\sum_{k=1}^{9} k^2 = \\frac{9 \\cdot 10 \\cdot 19}{6} = 285\n\\]\nMultiplying by 12:\n\\[\n12 \\sum_{k=1}^{9} k^2 = 12 \\cdot 285 = 3420\n\\]\n2. Sum of integers:\n\\[\n\\sum_{k=1}^{9} k = \\frac{9 \\cdot 10}{2} = 45\n\\]\nMultiplying by 6:\n\\[\n6 \\sum_{k=1}^{9} k = 6 \\cdot 45 = 270\n\\]\n3\n\n

Answer 2:\nTo find the value of the expression \\(2^3 - 1^3 + 4^3 - 3^3 + 6^3 - 5^3 + \\dots + 18^3 - 17^3\\), we can observe that it is a sum of differences of cubes in pairs. Let's break it down step by step.\n\nFirst, consider the general term in the sequence:\n\\[ (2k)^3 - (2k-1)^3 \\]\nfor \\( k = 1, 2, 3, \\ldots, 9 \\).\n\nWe need to simplify the expression \\((2k)^3 - (2k-1)^3\\). Using the identity for the difference of cubes, \\(a^3 - b^3 = (a-b)(a^2 + ab + b^2)\\), where \\(a = 2k\\) and \\(b = 2k-1\\), we get:\n\\[\n(2k)^3 - (2k-1)^3 = (2k - (2k-1)) \\left( (2k)^2 + (2k)(2k-1) + (2k-1)^2 \\right)\n\\]\nSimplifying inside the parentheses:\n\\[\n= 1 \\left( 4k^2 + 4k^2 - 2k + 4k^2 - 4k + 1 \\right)\n\\]\n= 4k^2 + 4k^2 - 2k + 4k^2 - 4k + 1\n= 12k^2 - 6k + 1\n\\]\n\nNow, we need to sum this expression from \\(k=1\\) to \\(k=9\\):\n\\[\n\\sum_{k=1}^{9} (12k^2 - 6k + 1)\n\\]\nWe can split this into three separate sums:\n\\[\n\\sum_{k=1}^{9} 12k^2 - \\sum_{k=1}^{9} 6k + \\sum_{k=1}^{9} 1\n\\]\nUsing the formulas for the sum of squares and the sum of the first \\(n\\) natural numbers:\n\\[\n\\sum_{k=1}^{n} k^2 = \\frac{n(n+1)(2n+1)}{6}\n\\]\n\\[\n\\sum_{k=1}^{n} k = \\frac{n(n+1)}{2}\n\\]\n\\[\n\\sum_{k=1}^{n} 1 = n\n\\]\nFor \\(n = 9\\):\n\\[\n\\sum_{k=1}^{9} k^2 = \\frac{9 \\cdot 10 \\cdot 19}{6} = 285\n\\]\n\\[\n\\sum_{k=1}^{9} k = \\frac{9 \\cdot 10}{2} = 45\n\\]\n\\[\n\\sum_{k=1}^{9} 1 = 9\n\\]\n\nSubstituting these values back into our expression:\n\\[\n12 \\sum_{k=1}^{9} k^2 - 6 \\sum\n\n

Answer 3:\nTo find the value of the expression \\(2^3 - 1^3 + 4^3 - 3^3 + 6^3 - 5^3 + \\dots + 18^3 - 17^3\\), we can observe that it consists of pairs of terms where each pair is of the form \\( (2n)^3 - (2n-1)^3 \\).\n\nFirst, let's simplify the general term in the sequence:\n\\[\n(2n)^3 - (2n-1)^3\n\\]\nUsing the binomial expansion for cubes, we have:\n\\[\na^3 - b^3 = (a - b)(a^2 + ab + b^2)\n\\]\nHere, \\(a = 2n\\) and \\(b = 2n-1\\). So,\n\\[\n(2n)^3 - (2n-1)^3 = (2n - (2n-1))((2n)^2 + (2n)(2n-1) + (2n-1)^2)\n\\]\nSimplifying inside the parentheses:\n\\[\n= 1 \\cdot ((2n)^2 + 2n(2n-1) + (2n-1)^2)\n\\]\n\\[\n= (4n^2 + 4n^2 - 2n + 4n^2 - 4n + 1)\n\\]\n\\[\n= 4n^2 + 4n^2 - 2n + 4n^2 - 4n + 1\n\\]\n\\[\n= 12n^2 - 6n + 1\n\\]\nNow, we need to sum these simplified terms from \\(n=1\\) to \\(n=9\\):\n\\[\n\\sum_{n=1}^{9} 12n^2 - 6n + 1\n\\]\nWe can split this into three separate sums:\n\\[\n\\sum_{n=1}^{9} 12n^2 - \\sum_{n=1}^{9} 6n + \\sum_{n=1}^{9} 1\n\\]\nCalculating each sum separately:\n1. Sum of squares:\n\\[\n\\sum_{n=1}^{9} n^2 = \\frac{9 \\cdot 10 \\cdot 19}{6} = 285\n\\]\nSo,\n\\[\n12 \\sum_{n=1}^{9} n^2 = 12 \\cdot 285 = 3420\n\\]\n2. Sum of integers:\n\\[\n\\sum_{n=1}^{9} n = \\frac{9 \\cdot 10}{2} = 45\n\\]\nSo,\n\\[\n6 \\sum_{n=1}^{9} n = 6 \\cdot 45 = 270\n\\]\n3. Sum of ones:\n\\[\n\\sum_{n\n\n

Answer 4:\nTo find the value of the expression \\(2^3 - 1^3 + 4^3 - 3^3 + 6^3 - 5^3 + \\dots + 18^3 - 17^3\\), we can observe that it is a sum of differences of cubes in pairs. Specifically, it can be written as:\n\\[\n(2^3 - 1^3) + (4^3 - 3^3) + (6^3 - 5^3) + \\dots + (18^3 - 17^3)\n\\]\nWe can use the identity for the difference of cubes, which states \\(a^3 - b^3 = (a - b)(a^2 + ab + b^2)\\). For each pair \\( (2k)^3 - (2k-1)^3 \\):\n\\[\n(2k)^3 - (2k-1)^3 = (2k - (2k-1)((2k)^2 + (2k)(2k-1) + (2k-1)^2)\n\\]\n\\[\n= 1 \\left(4k^2 + 4k^2 - 2k + 4k^2 - 4k + 1\\right)\n\\]\n\\[\n= 12k^2 - 6k + 1\n\\]\n\\[\n= 12k^2 - 6k + 1\n\\]\nSo, the expression simplifies to:\n\\[\n(2^3 - 1^3) + (4^3 - 3^3) + (6^3 - 5^3) + \\dots + (18^3 - 17^3) = (12 \\cdot 1^2 - 6 \\cdot 1 + 1) + (12 \\cdot 2^2 - 6 \\cdot 2 + 1) + \\dots + (12 \\cdot 9^2 - 6 \\cdot 9 + 1)\n\\]\nThis can be written as:\n\\[\n\\sum_{k=1}^{9} (12k^2 - 6k + 1)\n\\]\nWe can split this sum into three separate sums:\n\\[\n12 \\sum_{k=1}^{9} k^2 - 6 \\sum_{k=1}^{9} k + \\sum_{k=1}^{9} 1\n\\]\nUsing the formulas for the sum of the first \\(n\\) squares and the sum of the first \\(n\\) integers, we get:\n\\[\n\\sum_{k=1}^{9} k^2 = \\frac{9 \\cdot 10 \\cdot 19}{6} = 285\n\\]\n\\[\n\\sum_{k=1}^{9} k = \\frac{9 \\cdot 10}{2} = 45\n\\]\n\\[\n\\sum_{k=1}^{9} 1 = 9\n\\]\nSubstituting these val\n\n

Answer 5:\nTo solve the expression \\(2^3 - 1^3 + 4^3 - 3^3 + 6^3 - 5^3 + \\dots + 18^3 - 17^3\\), we can observe that it consists of pairs of terms where each pair is of the form \\( (2n)^3 - (2n-1)^3 \\).\n\nFirst, let's simplify the general term \\( (2n)^3 - (2n-1)^3 \\):\n\\[\n(2n)^3 - (2n-1)^3 = 8n^3 - (8n^3 - 12n^2 + 6n - 1) = 12n^2 - 6n + 1.\n\\]\n\nNow, we need to sum this expression for \\( n = 1 \\) to \\( n = 9 \\) (since the sequence goes up to 18 and 17, which correspond to \\( n = 1 \\) to \\( n = 9 \\)):\n\\[\n\\sum_{n=1}^{9} (12n^2 - 6n + 1).\n\\]\n\nWe can split this sum into three separate sums:\n\\[\n\\sum_{n=1}^{9} 12n^2 - \\sum_{n=1}^{9} 6n + \\sum_{n=1}^{9} 1.\n\\]\n\nUsing the formulas for the sum of squares and the sum of the first \\( n \\) natural numbers:\n\\[\n\\sum_{n=1}^{9} n^2 = \\frac{9 \\cdot 10 \\cdot 19}{6} = 285,\n\\]\n\\[\n\\sum_{n=1}^{9} n = \\frac{9 \\cdot 10}{2} = 45,\n\\]\n\\[\n\\sum_{n=1}^{9} 1 = 9.\n\\]\n\nSubstituting these values back in, we get:\n\\[\n12 \\sum_{n=1}^{9} n^2 - 6 \\sum_{n=1}^{9} n + \\sum_{n=1}^{9} 1 = 12 \\cdot 285 - .\n\n

Assistant: <think>Each proposed answer simplifies the general term and sums the series. The key is to recognize that the series is composed of pairs of cubes, and each pair simplifies to \\(12n^2 - 6n + 1\\). The sums of squares and integers from 1 to 9 are well-known and can be calculated directly. The total sum is the sum of these three parts: \\(12 \\cdot 285 - 6 \\cdot 45 + 9\\). Calculating each part: \\(12 \\cdot 285 = 3420\\), \\(6 \\cdot 45 = 270\\), and \\(9 = 9\\). Summing these: \\(3420 - 270 + 9 = 3159\\). All answers arrive at the same total, confirming the correctness of the calculation.</think> <answer>3159</answer>",

Figure 12: Example output when we cut off the last 10% of all answers to test whether SSA can synthesize. SSA SFT+RL 3B can synthesize and continue to get the final correct answer from AMC23 data. We can see the final answer 3159 does not appear anywhere in the provided solutions.

A conversation between User and Assistant. The user provide a question and some proposed answers. The Assistant first evaluate each answers individually,check whether each answer directly addresses the original question, assess the correctness of each answer based on logical reasoning, calculations, and accuracy relative to the question. After thorough evaluation, identify one correct answer. If the correct answer is not in the provided proposed answers, the Assistant will combine the correct answer with the proposed answers and provide the correct answer.The reasoning process and answer are enclosed within <think></think> and <answer></answer> tags, respectively, i.e., <think>reasoning process here</think> <answer>answer here</answer>. User: Question: A robe takes 2 bolts of blue fiber and half that much white fiber.  How many bolts in total does it take?\n\nBelow are the proposed answers:\n\n

Answer 1:\nTo determine the total number of bolts needed for the robe, we start by identifying the amount of each type of fiber required:\n\n- The robe requires 2 bolts of blue fiber.\n- It requires half as much white fiber as blue fiber, which means it needs \\( \\frac{1}{2} \\times 2 = 1 \\) bolt of white fiber.\n\nAdding these together gives us the total number of bolts:\n\n\\[ 2 \\text{ (blue)} + 1 \\text{ (white)} = 3 \\text{ bolts} \\]\n\n#### 3\n\n

Answer 2:\nTo determine the total number of bolts needed for the robe, we start by identifying the amount of each type of fiber required:\n\n1. The robe requires 2 bolts of blue fiber.\n2. The amount of white fiber required is half the amount of blue fiber, which is \\( \\frac{2}{2} = 1 \\) bolt.\n\nAdding these together gives the total number of bolts needed:\n\\[ 2 \\text{ (blue)} + 1 \\text{ (white)} = 3 \\text{ bolts} \\]\n\n#### 3\n\n

Answer 3:\nTo determine the total number of bolts of fiber needed for the robe, we start by identifying the amount of each type of fiber required.\n\n1. The robe requires 2 bolts of blue fiber.\n2. The amount of white fiber required is half that of the blue fiber. Therefore, the amount of white fiber needed is:\n  \\[\n  \\frac{2}{2} = 1 \\text{ bolt}\n  \\]\n\nNext, we add the bolts of blue fiber and white fiber together to find the total number of bolts needed:\n\\[\n2 \\text{ (blue)} + 1 \\text{ (white)} = 3 \\text{ bolts}\n\\]\n\nThus, the total number of bolts of fiber needed for the robe is:\n#### 3\n\n

Answer 4:\nA robe takes 2 bolts of blue fiber and half that amount in white fiber, which means it takes 1 bolt of white fiber (since half of 2 is 1). Therefore, the total number of bolts needed is 2 (blue) + 1 (white) = 3 bolts.\n\n#### 3\n\n

Answer 5:\nA robe takes 2 bolts of blue fiber and half that amount in white fiber, which means it takes 1 bolt of white fiber (since half of 2 is 1). \n\nIn total, the robe takes 2 bolts of blue fiber + 1 bolt of white fiber = 3 bolts.\n\n#### 3.

Assistant: <think>Each proposed answer correctly identifies that the robe requires 2 bolts of blue fiber and half that amount in white fiber, which is 1 bolt. The total bolts needed are then 2 (blue) + 1 (white) = 3 bolts. All answers arrive at the same conclusion through correct reasoning and calculations. Since all answers are consistent and correct, the most straightforward and clear answer is the one that explicitly states the total as 3 bolts, which is Answer 5. Therefore, the correct answer is 3 bolts.</think> <answer>3</answer>

Figure 13: SSA SFT Example. We can see there is a more clear reasoning on the response on the decision process.

A conversation between User and Assistant. The user provide a question and some proposed answers. The Assistant first evaluate each answers individually,check whether each answer directly addresses the original question, assess the correctness of each answer based on logical reasoning, calculations, and accuracy relative to the question. After thorough evaluation, identify one correct answer. If the correct answer is not in the provided proposed answers, the Assistant will combine the correct answer with the proposed answers and provide the correct answer.The reasoning process and answer are enclosed within <think></think> and <answer></answer> tags, respectively, i.e., <think>reasoning process here</think> <answer>answer here</answer>.

User: Question: Josh decides to try flipping a house. He buys a house for $80,000 and then puts in $50,000 in repairs. This increased the value of the house by 150%. How much profit did he make?\n\nBelow are the proposed answers:\n\n

Answer 1:\nFirst, we need to calculate the new value of the house after the repairs. The original cost of the house was $80,000, and Josh spent an additional $50,000 on repairs. Therefore, the total cost before considering the increase in value is:\n\n\\[ 80,000 + 50,000 = 130,000 \\]\n\nThe repairs increased the value of the house by 150%. To find the new value of the house, we need to calculate 150% of the original purchase price ($80,000) and add it to the original purchase price.\n\n\\[ 150\\% \\text{ of } 80,000 = 1.5 \\times 80,000 = 120,000 \\]\n\nAdding this to the original purchase price gives us the new value of the house:\n\n\\[ 80,000 + 120,000 = 200,000 \\]\n\nNow, we subtract the total amount Josh spent (the initial purchase price plus the repair costs) from the new value of the house to determine his profit:\n\n\\[ 200,000 - 130,000 = 70,000 \\]\n\nTherefore, Josh made a profit of $70,000.\n\n#### $70,000\n\n

Answer 2:\nFirst, we need to calculate the new value of the house after the repairs. The original cost of the house was $80,000, and Josh spent an additional $50,000 on repairs. Therefore, the total cost before considering the increase in value is:\n\n\\[ 80,000 + 50,000 = 130,000 \\]\n\nNext, we know that the repairs increased the value of the house by 150%. An increase of 150% means the value of the house became 250% (or 2.5 times) its original value before the repairs. So, the new value of the house is:\n\n\\[ 130,000 \\times 2.5 = 325,000 \\]\n\nNow, to find the profit, we subtract the total cost from the new value of the house:\n\n\\[ 325,000 - 130,000 = 195,000 \\]\n\nTherefore, the profit Josh made is:\n\n#### $195,000\n\n

Answer 3:\nFirst, let's calculate the new value of the house after the repairs. The original cost of the house was $80,000, and Josh spent an additional $50,000 on repairs. Therefore, the total cost before considering the increase in value is:\n\n\\[ 80,000 + 50,000 = 130,000 \\]\n\nNext, we need to determine the new value of the house after the 150% increase in value. A 150% increase means the value of the house becomes 250% (100% original value + 150% increase) of its pre-repair value. The pre-repair value of the house is $130,000, so the new value is:\n\n\\[ 130,000 \\times 2.5 = 325,000 \\]\n\nNow, to find the profit, we subtract the total cost from the new value of the house:\n\n\\[ 325,000 - 130,000 = 195,000 \\]\n\nTherefore, the profit Josh made is:\n\n#### $195,000\n\n

Answer 4:\nFirst, let's calculate the new value of the house after the repairs. The original purchase price of the house was $80,000, and Josh spent an additional $50,000 on repairs. Therefore, the total cost before considering the increase in value is:\n\n\\[ 80,000 + 50,000 = 130,000 \\]\n\nNext, we need to determine the new value of the house after it increased by 150%. An increase of 150% means the value of the house becomes 250% (or 2.5 times) its original purchase price. So, the new value of the house is:\n\n\\[ 80,000 \\times 2.5 = 200,000 \\]\n\nNow, we can find out how much profit Josh made by subtracting his total cost from the new value of the house:\n\n\\[ 200,000 - 130,000 = 70,000 \\]\n\nTherefore, the profit Josh made is:\n\n#### $70,000\n\n

Answer 5:\nFirst, we need to determine the new value of the house after the repairs. The original cost of the house was $80,000, and Josh spent an additional $50,000 on repairs. Therefore, the total cost before any increase in value is:\n\n\\[ 80,000 + 50,000 = 130,000 \\]\n\nThe problem states that the value of the house increased by 150% due to the repairs. An increase of 150% means the value of the house became 250% (or 2.5 times) its original cost after repairs. So, the new value of the house is:\n\n\\[ 130,000 \\times 2.5 = 325,000 \\]\n\nTo find the profit, we subtract the total cost from the new value of the house:\n\n\\[ 325,000 - 130,000 = 195,000 \\]\n\nTherefore, the profit Josh made is:\n\n#### $195,000.

Assistant: <think>reasoning process here</think> <answer>195000</answer>",

Figure 14: SSA RL Example. We can see the response reasoning process is just a copy of the instruction template. There are no meaningful content there

A conversation between User and Assistant. The user provide a question and some proposed answers. The Assistant answer the question based on the proposed answers. The answer is enclosed within <answer></answer> tag, i.e., <answer>answer here</answer>.

User: Question: Convert the point $(0,3)$ in rectangular coordinates to polar coordinates. Enter your answer in the form $(r,\theta),$ where $r > 0$ and $0 \le \theta < 2 \pi.$\n\nBelow are the proposed answers:\n\n

Answer 1:\nTo convert the point \\((0,3)\\) from rectangular coordinates to polar coordinates, we need to find the values of \\(r\\) and \\(\theta\\).\n\n1. **Calculate \\(r\\):**\n  The formula for \\(r\\) is:\n  \\[\n  r = \\sqrt{x^2 + y^2}\n  \\]\n  Substituting \\(x = 0\\) and \\(y = 3\\):\n  \\[\n  r = \\sqrt{0^2 + 3^2} = \\sqrt{9} = 3\n  \\]\n\n2. **Determine \\(\theta\\):**\n  The angle \\(\theta\\) can be found using the arctangent function:\n  \\[\n  \theta = \\tan^{-1}\\left(\\frac{y}{x}\\right)\n  \\]\n  However, since \\(x = 0\\) and \\(y = 3\\), this results in an undefined value for \\(\\tan^{-1}\\). Instead, we consider the position of the point in the coordinate plane. The point \\((0,3)\\) lies on the positive \\(y\\)-axis.\n\n  On the positive \\(y\\)-axis, the angle \\(\theta\\) is:\n  \\[\n  \theta = \\frac{\\pi}{2}\n  \\]\n\nThus, the polar coordinates of the point \\((0,3)\\) are:\n\\[\n(r, \theta) = (3, \\frac{\\pi}{2})\n\\]\n\n#### \\((3, \\frac{\\pi}{2})\\)\n\n

Answer 2:\nTo convert the point \\((0,3)\\) from rectangular coordinates to polar coordinates, we need to find the values of \\(r\\) and \\(\theta\\).\n\n1. **Calculate \\(r\\):**\n  The formula for \\(r\\) is:\n  \\[\n  r = \\sqrt{x^2 + y^2}\n  \\]\n  Substituting \\(x = 0\\) and \\(y = 3\\):\n  \\[\n  r = \\sqrt{0^2 + 3^2} = \\sqrt{9} = 3\n  \\]\n\n2. **Calculate \\(\theta\\):**\n  The formula for \\(\theta\\) is:\n  \\[\n  \theta = \\tan^{-1}\\left(\\frac{y}{x}\\right)\n  \\]\n  However, since \\(x = 0\\) and \\(y = 3\\), this point lies on the positive \\(y\\)-axis. Therefore, \\(\theta\\) should be \\(\\frac{\\pi}{2}\\).\n\nThus, the polar coordinates of the point \\((0,3)\\) are:\n\\[\n(r, \theta) = (3, \\frac{\\pi}{2})\n\\]\n\n#### \\((3, \\frac{\\pi}{2})\\)\n\n

Answer 3:\nTo convert the point \\((0,3)\\) from rectangular coordinates to polar coordinates, we need to find \\(r\\) and \\(\theta\\).\n\n1. **Calculate \\(r\\):**\n  The formula for \\(r\\) is:\n  \\[\n  r = \\sqrt{x^2 + y^2}\n  \\]\n  Substituting \\(x = 0\\) and \\(y = 3\\):\n  \\[\n  r = \\sqrt{0^2 + 3^2} = \\sqrt{9} = 3\n  \\]\n\n2. **Calculate \\(\theta\\):**\n  The formula for \\(\theta\\) is:\n  \\[\n  \theta = \\tan^{-1}\\left(\\frac{y}{x}\\right)\n  \\]\n  However, since \\(x = 0\\) and \\(y = 3\\), this results in an undefined value for \\(\\tan^{-1}\\). Instead, we consider the position of the point in the coordinate plane. The point \\((0,3)\\) lies on the positive \\(y\\)-axis.\n\n  On the \\(y\\)-axis, the angle \\(\theta\\) is \\(\\frac{\\pi}{2}\\) (or 90 degrees).\n\nTherefore, the polar coordinates of the point \\((0,3)\\) are:\n\\[\n(r, \theta) = (3, \\frac{\\pi}{2})\n\\]\n\n#### \\((3, \\frac{\\pi}{2})\\)\n\n

Answer 4:\nTo convert the point \\((0,3)\\) from rectangular coordinates to polar coordinates, we need to find the values of \\(r\\) and \\(\theta\\).\n\n1. **Calculate \\(r\\):**\n  The formula for \\(r\\) is given by:\n  \\[\n  r = \\sqrt{x^2 + y^2}\n  \\]\n  Substituting \\(x = 0\\) and \\(y = 3\\):\n  \\[\n  r = \\sqrt{0^2 + 3^2} = \\sqrt{9} = 3\n  \\]\n\n2. **Determine \\(\theta\\):**\n  The angle \\(\theta\\) can be found using the arctangent function:\n  \\[\n  \theta = \\tan^{-1}\\left(\\frac{y}{x}\\right)\n  \\]\n  However, since \\(x = 0\\) and \\(y = 3\\), this directly places the point on the positive \\(y\\)-axis. Therefore, the angle \\(\theta\\) is:\n  \\[\n  \theta = \\frac{\\pi}{2}\n  \\]\n\nThus, the polar coordinates of the point \\((0,3)\\) are:\n\\[\n(r, \theta) = (3, \\frac{\\pi}{2})\n\\]\n\n#### \\((3, \\frac{\\pi}{2})\\)\n\n

Answer 5:\nTo convert the point \\((0, 3)\\) from rectangular coordinates to polar coordinates, we need to find \\(r\\) and \\(\theta\\).\n\n1. **Calculate \\(r\\):**\n  The formula for \\(r\\) is:\n  \\[\n  r = \\sqrt{x^2 + y^2}\n  \\]\n  Substituting \\(x = 0\\) and \\(y = 3\\):\n  \\[\n  r = \\sqrt{0^2 + 3^2} = \\sqrt{9} = 3\n  \\]\n\n2. **Calculate \\(\theta\\):**\n  The formula for \\(\theta\\) is:\n  \\[\n  \theta = \\tan^{-1}\\left(\\frac{y}{x}\\right)\n  \\]\n  However, since \\(x = 0\\), \\(\\tan^{-1}\\left(\\frac{3}{0}\\right)\\) is undefined. This indicates that the point lies on the positive \\(y\\)-axis. Therefore, \\(\theta\\) must be \\(\\frac{\\pi}{2}\\).\n\nPutting it all together, the polar coordinates are:\n\\[\n(r, \theta) = (3, \\frac{\\pi}{2})\n\\]\n\n#### \\((3, \\frac{\\pi}{2})\\).

Assistant: <answer>3, pi/2</answer>

Figure 15: SSA No Think Example. We can see the Assistant output answer directly

# D  THE USE OF LARGE LANGUAGE MODELS (LLMs)

We use LLM to polish our writing. In addition, LLM also assist our coding and visualization.

