# OpenReview forum: "Learning to Reason Across Parallel Samples for LLM Reasoning"
_ICLR.cc/2026/Conference — Submitted to ICLR 2026_

### Official Review · Reviewer_3tVa · 2025-10-17

**Soundness:** 2
**Presentation:** 3
**Contribution:** 2
**Rating:** 0
**Confidence:** 4

**Summary:**

The paper proposes Sample Set Aggregator (SSA): a small LLM reads and concatenates K parallel sampled answers, and uses GRPO reinforcement learning to directly optimize the “final correctness rate.” SSA decouples “generation” and “aggregation,” serving as an efficient answer combiner for black-box large models. On benchmarks such as GSM8K, MATH, AIME24, AMC23, and Olympiad, SSA consistently outperforms majority voting and various PRM/ORM baselines; the 3B version achieves a better efficiency–accuracy trade-off and shows good generalization across model sizes and families.

**Strengths:**

1. The paper is well written and easy to follow.
2. The experiments are comprehensive and strongly support the authors’ claims.

**Weaknesses:**

1. The motivation is not sufficiently novel. According to Lines 56–65, the method simply combines two test-time scaling approaches. It neither analyzes which limitations of existing methods this addresses, nor does it offer conceptual insight; the contribution reads more as engineering practice and provides limited guidance for future research.
2. There is a lack of comparison with prior work. Given many studies train verifiers to select best-of-N [1, 2], why is the training approach in Section 3 more effective than these? The experimental section also lacks such comparisons.

Considering the work is overly trivial and provides limited research insight, I recommend rejection.

[1] Mudgal et al., Controlled decoding from language models. ICML 2024.

[2] Huang et al., Is best-of-n the best of them? coverage, scaling, and optimality in inference-time alignment. Arxiv.

**Questions:**

1. Given that many scenarios lack training data, how can your method be applied in such settings?
2. Additional experiments on long-CoT models (e.g., DeepSeek-R1-Distill-Llama, Qwen3) and broader datasets (e.g., MMLU-Pro, GPQA) would better substantiate the method’s effectiveness.

---

> ### Author Response · Authors · 2025-11-22
>
> >The motivation is not sufficiently novel. According to Lines 56–65, the method simply combines two test-time scaling approaches.
>
> We respectfully disagree. SSA is optimizing over the output distribution, rather than updating base model weights. Prior RL-based scaling methods change the generator’s parameters to directly reshape the output distribution. SSA instead treats the base model as fixed, samples from its current distribution, and trains a small aggregator that directly maps a set of candidate solutions to a final answer. Best to our knowledge, SSA is a new method that bridges parallel and sequential test-time compute.
>
> >There is a lack of comparison with prior work. Given many studies train verifiers to select best-of-N [1, 2], why is the training approach in Section 3 more effective than these?
>
> For the comparison with prior best-of-N, we already compare against strong best-of-N style baselines that train verifiers to select the best candidate, including PRM, ORM, USC, and RLVR-tuned sequential models (SimpleRL). These appear both in the related work and in the experimental section. The cited works [1, 2] are closely relevant but not themselves verifier architectures. [1] study controlled decoding and inference-time alignment. [2] provide a theoretical analysis of best-of-N schemes (coverage, scaling, optimality).

---

> > ### Comment · Reviewer_3tVa · 2025-11-23
> >
> > Thank you for your response. However, I still think that the work is an incremental engineering work, which lacks intrinsic inspiration and does not meet the meeting standards of ICLR, so I decided to keep the score.

---

### Official Review · Reviewer_wiEM · 2025-10-29

**Soundness:** 3
**Presentation:** 3
**Contribution:** 2
**Rating:** 4
**Confidence:** 3

**Summary:**

The paper proposes an test-time scaling aggregation method that involves training an LLM (Sample Set Aggregator, SSA) to aggregate multiple parallel responses from a base model (concatenated together as input) and output the final response. The SSA is trained via reinforcement learning (GRPO) and has been tested empirically on various math reasoning tasks and two LLM model families. The paper demonstrates empirically how a smaller SSA model (3B model) can surpass the performance fo a much larger 72B process reward model (PRM).

**Strengths:**

- The paper tackles an important problem of test-time scaling and how search/optimization/aggregation methods could be used to improve model performance, including when the base model is a blackbox/API-only model.
- The paper is relatively well written, with clear structure and presentation
- The paper presents empirical results indicating that their proposed SSA model could match the performance or even outperform larger PRMs, including a 3B SSA model matching/outperforming the Qwen 72B PRM model.
- The authors have provided relatively extensive experimental results, including in the appendix.

**Weaknesses:**

- The paper should include more detailed comparisons and discussions with the LLM ensemble literature where a base model could be queried in parallel with various different reasoning/role prompts with these responses subsequently aggregated [1-3]. These methods bear similarity in that K parallel candidates are also drawn from a frozen base model before being aggregated, with the aggregation possibly also being an LLM fed with the various candidate responses.

- The paper's claim in proposing a method that is "learning to reason" seems unjustified, especially based on the empirical results provided. As the authors discussed in Sec 6.2, the SSA appears to largely be playing the role of a response selector rather than be a model that actually reasons over the candidate responses -- the model only output placeholder thinking tokens, and the authors noted minor performance degradation without explicit thinking. Appendix Table 9 and 10 also supports this perspective. The authors should consider repositioning the paper to clearly reflect this.

- The paper could benefit from further analysis of how the training data distribution affects the performance and trends of SSA, such as whether removing (or only including) 5/5 or 0/5 (very easy or very tough) questions from the training data would affect the model's 'selection' vs 'synthesis' behavior and overall performance.

- The SSA model seems to not be able to generalize well beyond math tasks, which it has been trained on. Table 11 in the appendix shows inconsistent performance of SSA over majority vote (the only baseline presented with just k=5). The authors could consider providing further analysis to describe the limitations of generalizability of the method.

- The baseline USC, which is the vanilla LLM aggregator approach or SSA without training, seems to have a prompt that is biasing it towards performing majority vote. Fig 10 in the appendix shows that compared to the SSA prompt, the USC prompt has been modified to instruct the model to answer "based on majority consensus". The authors should explain why this is justified.

- The scalability of this approach seems to be a potential issue, especially if there are responses with longer reasoning traces or solutions. While the authors proposed the multi-stage SSA approach, this might lead to increased inference cost and degrade performance for other datasets with longer responses. Appendix A.4 also seems to hint that training performance may also be limited by context window, but more analysis would be needed to check

[1] Du et al, Improving factuality and Reasoning in Language Models through Multiagent Debate

[2] Hu et al, Dipper: Diversity in Prompts for Producing Large Language Model Ensembles in Reasoning Tasks

[3] Chen et al, AgentVerse: Facilitating Multi-Agent Collaboration and Exploring Emergent Behaviors.

**Questions:**

Please see the weakness section for questions.

---

> ### Author Response · Authors · 2025-11-22
>
> We thank the reviewer for the detailed and constructive feedback. Below we address each point in turn.
> > More discussions with the LLM ensemble literature
>
> We appreciate the pointers and will add discussions on multiagent debate. Conceptually, those works and SSA both use multiple models’ responses to enhance the final answer. Those works’ focuses are prompt and role diversity, and all three works are training-free methods to ensemble different answers. In contrast, SSA trains a small LLM which takes model samples to improve the final answer.
>
> >The paper's claim in proposing a method that is "learning to reason" seems unjustified
>
> We agree that SSA+RL often behaves like a selector rather than reasoner, and appreciate the reviewer's detailed reading of our model analysis.
>
> We’d like to highlight an analysis in Appendix A.7 which partially addresses your concern. It provides evidence that SSA does more than simple selection. When we truncate away the last 10% of each chain-of-thought (including the removal of the final answer tokens), a purely “selector-style” model would be severely handicapped. Under this truncation, SSA+RL (with truncated answers) still achieves 34.5% average accuracy, and SSA+RL+SFT (with truncated answers) reaches 47.09%, compared to 56.13% for the original SSA+RL (with full answers). This gap shows that SSA+RL can partially reconstruct or synthesize answers from incomplete rationales rather than just picking the best visible final answer. However SSA+SFT+RL in particular is more robust.
>
> That said, we agree that framing SSA’s behavior as “reasoning” might not be most accurate, given how word reasoning is used in recent literature. We will update the title as “learning to use parallel samples for llm reasoning” and discuss this point. We believe future follow up work can focus on building an aggregator that goes substantially beyond selector. We will add discussion in the paper.
>
> >The paper could benefit from further analysis of how the training data distribution affects the performance and trends of SSA
>
> We report training-set statistics in Appendix A.11, including the distribution of sets with 0/5, 1/5, …, 5/5 correct answers. Intuitively, 5/5 sets correspond to very easy questions, while 0/5 sets are extremely hard. Training a selector with only a certain correct answer rate will likely harm performances, as at test time it’ll be exposed to various correct answer rates.

---

> > ### Author Response · Authors · 2025-11-22
> >
> > >The SSA model seems to not be able to generalize well beyond math tasks, which it has been trained on.
> >
> > We agree that the SSA+RL is tightly tuned to math and does not generalize well beyond that domain even though it can generalize across model families and sizes. Our main focus of the paper is in-distribution test-time scaling for math reasoning. At the same time, Table 11 shows more interesting insights. SSA+RL can overfit to the math distribution and underperform the majority vote on some out-of-domain tasks. SSA+SFT and SSA+SFT+RL exhibit more robust behavior and sometimes improve over majority vote. We will update the text around Table 11 and in Appendix A.8 to reflect this.
> >
> > >The baseline USC, which is the vanilla LLM aggregator approach or SSA without training, seems to have a prompt that is biasing it towards performing a majority vote. The authors should explain why you modify the original USC prompt.
> >
> > We would like to clarify that our version of USC prompt outperforms the original USC prompt. In the Table below, we show the performance of USC with three prompt settings 1) SSA prompt (the prompt used for training) 2) original USC prompt and 3) Adapted USC prompt used in our original evaluation. As shown in the Table below, using our adapted USC prompt leads to best performance (i.e., we tweaked the prompt a little bit to improve the baseline performance).
> >
> > | **Method** | **GSM8K** | **MATH** | **AIME24** | **AMC23** | **Olympiad** | **Avg** |
> > |------------|-----------|----------|------------|-----------|--------------|---------|
> > | USC / SSA Prompt | 75.1% (+0.12%) | 53.4% (-7.8%) | 3.33% (-3.3%) | 40.00% (-7.5%) | 22.6% (-6.2%) | 38.89% (-4.94%) |
> > | Original USC Prompt | 81.12% (+6.14%) | 56.8% (-4.4%) | 3.33% (-3.3%) | 45.00% (-2.5%) | 25.37% (-3.41%) | 42.32% (-1.51%) |
> > | Adapted USC prompt (ours) | 74.98% | 61.20% | 6.67% | 47.50% | 28.78% | 43.83% |
> >
> > >The scalability of this approach seems to be a potential issue
> >
> > We acknowledge that as context length increases, our inference costs (sequence of parallel samples), when used naively, can be costly. However, in many test-time scaling scenarios, large-scale sampling of parallel samples themselves are costly, making light-weight SSA cost modest in comparison. We report Table 8 and Table 12 the actual wall-clock overhead from adding SSA on top of the base generator. Across benchmarks, the extra cost from the 3B SSA forward pass is modest relative to the cost of generating $K$ answers from a 7B/14B/32B base model. Multi-stage SSA does add some inference overhead, but it also respects context limits inherent in LLMs.
> >
> >
> >
> >
> >
> > [1] Chen, Xinyun, et al. "Universal self-consistency for large language model generation." arXiv preprint arXiv:2311.17311 (2023).

---

### Official Review · Reviewer_hysi · 2025-10-31

**Soundness:** 3
**Presentation:** 3
**Contribution:** 2
**Rating:** 2
**Confidence:** 4

**Summary:**

This paper proposes Sample Set Aggregator (SSA) that trains smaller models using RL to aggregate multiple sampled answers from a larger base model. Experiments on 5 math reasoning benchmarks show that SSA outperforms majority votes and re-ranking methods that rely on larger reward models (including ORM and PRM). This paper also offers results on generalization across sample size as well as model families and sizes.

**Strengths:**

- The core idea is simple yet effective, particularly for in-domain math reasoning tasks.

- Experiments across 5 datasets demonstrate that the RL-trained, smaller SSA model can outperform larger ORM and PRM models on math reasoning benchmarks.

**Weaknesses:**

- The method is conceptually straightforward and lacks strong technical novelty, and the empirical gains are not substantial enough to offset this limitation.

- Improvements are mostly limited to math reasoning, where prior ORM and PRM methods have shown broader generalization across diverse domains.

- As shown in Figure 2, performance does not generalize well when increasing the number of input samples, and in Table 2, the results using Llama to produce answer input are also only marginally better than baselines.

- The comparison with sequential scaling in Section 5.3 uses a smaller model against much larger models; a fairer comparison with models of similar size would strengthen the claims.

**Questions:**

- A possible generalization experiment could involve aggregating answers from multiple base models, can be within the same family, to reduce the need for repeated sampling from large models. Have the authors explored this direction?

---

> ### Author Response · Authors · 2025-11-22
>
> We thank the reviewer for the detailed feedback and address the main points below.
> > The method is conceptually straightforward and lacks strong technical novelty
>
> We respectfully disagree that conceptual simplicity is a weakness. We present the first work to bridge the parallel and sequential test time compute method. In the introduction, we clarified this concept that we are optimizing on sampled output distribution directly instead of updating LLM’s weights to change the output distribution. We believe it is conceptually novel. This gives us a clean bridge between parallel sampling (cheap but independent) and sequential scaling (expensive but powerful). To our knowledge, SSA is the first method that explicitly frames test-time compute in terms of “operating on sample sets” rather than “operating on model weights”. In addition, we see a large empirical gain (+2.73%) on average even compared to a 24 times larger model.
>
> >Improvements are mostly limited to math reasoning
>
> Our primary goal is in distribution output distribution sharpening for math reasoning, so we followed the standard to train and evaluate on math benchmarks similar to many prior works [1,2, 3]. In that setting, RL-based SSA is a natural choice and performs strongly. That said, SSA is not restricted to math. In Table 11, we show that the SSA SFT variant generalizes better to adjacent domains. We view broader cross domain generalization as complementary future work rather than a core claim of this submission.
>
> > As shown in Figure 2, performance does not generalize well when increasing the number of input samples
>
> Fig. 2 does not indicate poor generalization in $K$. From $K=5$ to $K=15$, SSA consistently improves over the baseline and maintains a similar gap. Beyond that, we also study larger $K$ in Section 5.4, scaling up to 128 samples. There we additionally introduce hierarchical aggregation, where SSA aggregates in stages rather than over a single very long concatenation. This further mitigates any potential degradation with large $K$ when answers are out of LLM effective context.
> On the LLaMA-generated candidates (Table 2), the gains are smaller than in the main Qwen setting, but SSA still improves over majority vote and other baselines.  Our goal in this experiment was to test out of training distribution generators (different model families), and SSA continues to give consistent, non-negative improvements there.
>
> > The comparison with sequential scaling in Section 5.3 uses a smaller model against much larger models; a fairer comparison with models of similar size would strengthen the claims.
>
> We would like to clarify potential misunderstandings. In section 5.3, our intention is to compare two ways of using test-time compute with large base models (7B - 32B): (1) Sequential RL scaling: RL-tune the 7B/14B/32B model itself. (2) SSA scaling: Keep the 7B/14B/32B models frozen, sample from them in parallel, and aggregate with a fixed 3B SSA. In both cases, we control the answer generating model to be of the same size. This is actually an unfair comparison to our method, as our method RL-tunes a much compact model. Under this lens comparing RL-tuned 3B SSA to RL-tuned {7B, 14B, 32B} reflects the intended question: how close can we get to fully RL-tuned large models while only training a much smaller auxiliary model once? Our results shows SSA closes the gaps with respect to RL-tuned large base models despite using much few compute.
>
> If we instead compare SSA to an RL-tuned similar scaled model while still using the model as generators for SSA, this will make the gap between SSA and sequential scaling larger. We summarize below the reported results for 1.5B GRPO base models from [1], versus our 7B generator with 1.5B SSA. Note that [1] does not report a 3B mode.
> | **Method** | **GSM8K** | **MATH** | **AIME24** | **AMC23** | **Olympiad** | **Avg** |
> |------------|-----------|----------|------------|-----------|--------------|---------|
> | 1.5B GRPO [1]  (%) | 74.4  | 59.0 | 6.7  | 35.0 | 21.0 | 39.22 |
> | 7B + 1.5B SSA (%) | 92.49 | 76.60  | 10.00  | 52.50 | 38.72  | 54.06 |

---

> > ### Author Response · Authors · 2025-11-22
> >
> > >A possible generalization experiment could involve aggregating answers from multiple base models
> >
> > We agree this is an interesting direction! We have run an additional experiment where SSA aggregates samples from multiple base models. Specifically, we draw 2 samples from Qwen-2.5-7B, 2 samples from 14B, and 2 samples from 32B (for a total of 6 answers), concatenate them, and feed them into the same SSA aggregator. We report the result in this setting below, comparing USC, majority vote, and our method.
> >
> > | **Method** | **GSM8K** | **MATH** | **AIME24** | **AMC23** | **Olympiad** | **Avg** |
> > |------------|-----------|----------|------------|-----------|--------------|---------|
> > | USC 7B| 88.1% | 72.2% | 6.67% | 60% | 37.54%% |52.9%|
> > | Majority| 95.98% | 75% | 13.33% | 55% | 38.28% |55.52%|
> > | SSA | 95.91% | 81.80% | 16.67% | 57.50% | 44.51% |59.29%|
> >
> >
> > This suggests that SSA can effectively exploit heterogeneous candidate sets and that aggregating multiple base models is a promising extension.
> >
> > [1] Zeng, Weihao, et al. "Simplerl-zoo: Investigating and taming zero reinforcement learning for open base models in the wild." arXiv preprint arXiv:2503.18892 (2025).
> >
> > [2] Zhang, Zhenru, et al. "The lessons of developing process reward models in mathematical reasoning." arXiv preprint arXiv:2501.07301 (2025).
> >
> > [3] Yue, Yang, et al. "Does reinforcement learning really incentivize reasoning capacity in llms beyond the base model?." arXiv preprint arXiv:2504.13837 (2025).

---

> > > ### Comment · Reviewer_hysi · 2025-11-28
> > >
> > > I want to thank the authors for taking the time to respond to my review. I appreciate the new results by combining multiple models of different sizes, though from the same family. I'm not yet convinced by the claimed novelty and experiments within math reasoning. I'll maintain my score.

---

### Official Review · Reviewer_FYUV · 2025-10-31

**Soundness:** 3
**Presentation:** 4
**Contribution:** 3
**Rating:** 6
**Confidence:** 4

**Summary:**

This paper presents SSA, a technique that combines parallel and sequential reasoning techniques to help LLMs decide best answer from LLM responses. The key component: the SSA model get parallel responses from the LLM, and reason on top of the concatenated results. The model can be trained in two ways: use RL to reward generated response or use supervised FT based on GPT-4.1 nano. The results show that (1) SSA is a lightweight approach that effectively outperform parallel scoring and getting close to sequential scaling with RL, and (2) using FT to train SSA model performs better than RL based approach.

**Strengths:**

- simple but very clean problem formulation. The lightweight test-time scaling combines strengths of parallel sampling (cheap) with ability to reason across solutions in sequential.
- comparison of training algorithms (SFT vs RL) for training SSA model
- evaluation showing generalization beyond training from model with different responses

**Weaknesses:**

- The comparison with reranker model is not quite clear to me, which also reason on top of parallel sampling results from the LLM responses.
- The paper primarily focuses on smaller/lightweight model. It's ideal to understand whether SSA model performance tops at sequential reasoning, by ablation on SSA model size and SFT reference model responses (e.g., using larger reasoning model). This is particularly interesting especially that smaller model may not be able to leverage large reasoning model's samples to generate highquality results.

**Questions:**

- (important) Please elaborate more on how reranker / PRM model approach compares with SSA, as both leverage parallel samples for reasoning.
- provide more discussions about how model size / reference model size in SFT affect SSA performance.

---

> ### Author Response · Authors · 2025-11-22
>
> We thank the reviewer for the thoughtful and constructive feedback.
>
> > Comparison with reranker:
>
> All reranker models (e.g., PRM, ORM) consider each candidate solution on its own: given a question and one chain-of-thought, they output a scalar score, and the final answer is chosen by argmax over these independent scores. In contrast, our model reads all $K$ answers jointly in a single context then generates the final answer.
>
> > SSA model size vs. Response model size:
>
> This is a good point!
>
> We have varied the size of SSA models (Fig. 2 and Tab. 4, 0.5B, 1.5B, and 3B SSA models). We see modest gains for using larger SSA models, but we do not think the headroom is significant, given 1.5B and 3B SSA model performance is not very different.
>
> In Fig. 3 we compare 3B SSA on top of Qwen 2.5 7B/14B/32B to fully RL trained sequential models of the same sizes from SimpleRL-Zoo under matched test-time compute. SSA comes close but does not surpass the best sequential RL models, which is expected since those methods directly tune the 7B/14B/32B base models.  SSA offers a lighter-weight alternative to full RL training. We only RL-train a 3B model once, yet can plug it on top of multiple base scales. We believe the practical “ceiling’’ is indeed full RL on the base model since it has direct output distribution access without sampling to approximate it. Our value is in approaching that ceiling with far smaller training and deployment costs.
>
> For the reference model size, we compared that in Section 5.3. We compared 3B SSA on top of the 7B, 14B, and 32B models without retraining. We see the performance improves monotonically as the base model scale increases. We will make this discussion explicit in Sec. 5.3.

---

### Meta-Review · Area_Chair_SCXR · 2026-01-05

**Summary:**

The reviewers expressed the following concerns:

C1. The paper lacks substantial technical novelty, constituting a combination of sequential and parallel scaling. Although this is a novel concept, the combination does not involve any technical development beyond standard RL training or supervised fine-tuning of a lightweight aggregator.

C2. The main experimental analysis and demonstrated gains are limited to math reasoning tasks and the model is trained using two of the evaluation datasets. Results in the appendix suggest that the approach does not generalize to other reasoning modalities.

C3. The performance does not generalize well when increasing the number input samples and the results when using Llama to produce the answer input are only marginally better than baselines.

C4. Scalability is an unaddressed issue.

C5. The paper focuses on smaller, lightweight models.

C6. The comparison with reranker models is unclear.

C7. There is very limited analysis of the impact of the training distribution on the performance.

**Reviewer Concerns:**

C1: While the response provided a rebuttal to concern C1 (expressed in different ways by two reviewers), it focused on the criticism of conceptual simplicity rather than clearly identifying the meaningful technical contribution. The reviews did not dispute the conceptual novelty. The concept is novel, but the technical contribution after that concept is proposed does appear limited for an ICLR paper, representing a standard application of RL or fine-tuning. It is not clear that the experiments are sufficiently extensive or insightful to compensate for the lack of methodological innovation.

C2. The response acknowledges that the analysis and performance improvement is limited to mathematical reasoning, with some limited experiments in the appendix for other types of reasoning (that don't clearly identify any advantage). Given the lack of a significant methodological technical contribution, one could reasonably expect a considerably more thorough experimental investigation of this, delving into why the approach does not achieve success for other reasoning tasks.

C3. The response addresses criticism C3.

C4. While the response does discuss criticism C4, the paper does not provide clear evidence that the technique can scale, and this remains a valid concern.

C5. This concern is satisfactorily addressed.

C6. This concern is addressed very briefly in the response, but appears to be a satisfactory clarification.

C7. This concern remains unaddressed. The response states that changing the training distribution "will likely harm performances, as at test time it’ll be exposed to various correct answer rates" but no experimental evidence is provided. This is a weakness of the experimental investigation. Conducting an analysis using a single training dataset raises questions about how well the technique generalizes if trained on other datasets.

**Reviewer Scores:**

Reviewer FYUV. POSSIBLY INCREASED SCORE. The response appears to address most of the criticisms raised by this reviewer, so there is a reasonable chance that the score would be increased. It was already a 6.

Reviewer hysi. NO CHANGE (Score 2). The reviewer acknowledged the response, but stated that the score remained the same. This reviewer expressed concerns C1 and C2, which are two of the concerns that the response fails to address convincingly.

Reviewer wiEM. VERY LIKELY NO CHANGE (Score 4). This reviewer identified multiple weaknesses. While some of these are successfully addressed in the response, concerns such as C2, C4, and C7, which were raised by this reviewer in some form, all remain unconvincingly addressed.

Reviewer 3tVa. NO CHANGE (Score 0). I would recommend placing little weight on this review. The criticisms are not clearly expressed and in some cases appear to miss the point of the paper. Having said that, the response fails to directly address the two questions raised by the review.

---

### Decision · Program_Chairs · 2026-01-26

Reject